# Causal Dependence Plots

**Joshua R. Loftus**
Department of Statistics
London School of Economics
London, England, UK
j.r.loftus@lse.ac.uk

**Lucius E. J. Bynum**
Center for Data Science
New York University
New York, NY, USA
lucius@nyu.edu

**Sakina Hansen**
Department of Statistics
London School of Economics
London, England, UK
s.a.hansen1@lse.ac.uk

## Abstract

To use artificial intelligence and machine learning models wisely we must understand how they interact with the world, including how they depend causally on data inputs. In this work we develop Causal Dependence Plots (CDPs) to visualize how a model's predicted outcome depends on changes in a given predictor *along with consequent causal changes in other predictor variables*. Crucially, this differs from standard methods based on independence or holding other predictors constant, such as regression coefficients or Partial Dependence Plots (PDPs). Our explanatory framework generalizes PDPs, including them as a special case, as well as a variety of other interpretive plots that show, for example, the total, direct, and indirect effects of causal mediation. We demonstrate with simulations and real data experiments how CDPs can be combined in a modular way with methods for causal learning or sensitivity analysis. Since people often think causally about input-output dependence, CDPs can be powerful tools in the xAI or interpretable machine learning toolkit and contribute to applications like scientific machine learning and algorithmic fairness.

## 1 Introduction

This paper develops Causal Dependence Plots (CDPs) to visualize relationships between input variables and a predicted outcome. Motivated by explaining or interpreting AI or machine learning models [8, 16, 17, 36], for simplicity we consider supervised learning, i.e. regression or classification. We also focus on the model-agnostic or "black-box" setting, where the interpreter can query the model but not access its internal structure. Model-agnostic interpretation methods are functionally limited to observing how the model responds to variation in the inputs. While this initial application forms our practical motivation, we emphasize that CDPs are more general.

Simple explanations that focus on one input variable at a time can be powerful tools for human understanding. However, just as with the interpretation of linear regression model coefficients, these simple relationships can be misleading. When varying one input variable, *we must make some choice about what values to use for the other inputs*. CDPs make this choice using an explicit causal model, and to our knowledge this is the first work that does so. We compare CDPs to other state-of-the-art, non-causal explanation methods like the Partial Dependence Plot (PDP) [12], Individual Conditional Expectation (ICE) [14], Accumulated Local Effect (ALE) [3], and Shapley Additive Explanation (SHAP) feature plot [33]. Explanation methods may respect existing causal dependencies between predictors or break them.

**Problem statement.** If there are causal relationships between predictors but our visualization, interpretation, or explanation method does not respect them the resulting model explanation may be irrelevant or misleading [37, 54]. Such explanations could lead to incorrect decisions for regulating or aligning algorithmic systems, sub-optimal allocations of resources based on model predictions, a breakdown between human feedback and reinforcement learning systems, or other forms of error and

38th Conference on Neural Information Processing Systems (NeurIPS 2024).

harm. In scientific machine learning—where explanations can be used to generate hypotheses for follow-up investigation—a flawed interpretation may support spurious hypotheses. For these reasons, *the causal validity of model explanations should be a top priority*.

**High level proposal.** We wish to interpret or explain a given supervised machine learning model $\hat{f}(\mathbf{x})$ with $p$ input features $\mathbf{x} = (x_1, \ldots, x_p)$. Specifically, we want to understand how the predictions $\hat{y} = \hat{f}(\mathbf{x})$ of this model depend on feature $x_j$ for a given $j$, $1 \leq j \leq p$. PDPs and ICE plots do this by varying $x_j$ and holding the other features $\mathbf{x}_{\backslash j}$ constant, where $\mathbf{x}_{\backslash j}$ is the $(p-1)$-tuple containing all features except for $x_j$. This implicitly assumes independence between $x_j$ and the other inputs. Our method replaces this independence assumption with an Explanatory Causal Model (ECM) for the input features. This auxiliary ECM is a tool we use to help explain $\hat{f}$, it determines how other inputs $\mathbf{x}_{\backslash j}$ vary when $x_j$ is changed.

**CDP pseudo-algorithm.** To construct a CDP showing how $\hat{f}(\mathbf{x})$ depends on $x_j$, a user specifies an ECM $\mathcal{M}$ containing the predictors $\mathbf{x}$, and an intervention $I(x_j)$ in $\mathcal{M}$. The intervention changes $x_j$, and may change other features if they are caused by $x_j$ in $\mathcal{M}$. The type of intervention is chosen based on the type of causal explanation desired, with several example options demonstrated later. An explanatory dataset $\mathcal{D} = \{\mathbf{x}_i : i = 1, \ldots, n\}$ can be given or, if unavailable, generated by $\mathcal{M}$. Note that we use the notational convention where $i$ indexes observations or examples while $j$ indexes features. The horizontal axis of the plot is specified by a grid $\{\tilde{x}_{j,k} : k = 1, \ldots, K\}$ of possible values for $x_j$, with $k$ indexing grid points. For each observation $\mathbf{x}_i$ in $\mathcal{D}$, and at each grid point $\tilde{x}_{j,k}$:

1. Use the ECM to simulate counterfactual values $\mathbf{x}_{i,k}^*$ for all features of observation $i$ under the intervention $I(\tilde{x}_{j,k})$.
2. Input counterfactual features to the prediction function $\hat{f}$, and store the resulting counterfactual prediction $\hat{y}_{i,k}^* = \hat{f}(\mathbf{x}_{i,k}^*)$ in an array indexed by $(i, k)$.

For each observation $i$ in $\mathcal{D}$, construct the individual counterfactual prediction curve $(\tilde{x}_{j,k}, \hat{y}_{i,k}^*)$ by connecting points that are adjacent on the plot grid. Plot the empirical average of these curves, which is the main output of the CDP. The individual curves can be shown or suppressed as desired. *The resulting CDP shows how the model's predictions $\hat{y}$ causally depend on $x_j$ when this predictor is varied by the intervention $I(x_j)$ in ECM $\mathcal{M}$.*

**PDP and ICE algorithm.** Start with the notation and setup as above but without any ECM or intervention. For each observation $\mathbf{x}_i$ in $\mathcal{D}$, and at each grid point $\tilde{x}_{j,k}$:

1. Define $\mathbf{x}_{i,k}'$ by setting feature $x_{ij}$ to the grid point $\tilde{x}_{j,k}$ and keeping other features $\mathbf{x}_{\backslash j}$ fixed at their original values in $\mathbf{x}_i$ from $\mathcal{D}$, that is

$$\mathbf{x}_{i,k}' := (x_{i1}, \ldots, x_{ij} \leftarrow \tilde{x}_{j,k}, \ldots, x_{ip}). \tag{1}$$

2. Compute prediction $\hat{y}_{i,k}' = \hat{f}(\mathbf{x}_{i,k}')$ and store in an array indexed by $(i, k)$.

Plotting $(\tilde{x}_{j,k}, \hat{y}_{i,k}')$ generates an ICE curve for each $i$, and the empirical average of these is the PDP.

**Motivating example.** Consider a model for parental income $P$, school funding $F$, and graduates' average starting salary $S$, with ECM shown in the bottom row of Figure 1. In the top row, the ECM functions are plotted in the left panel, and the remaining panels show visual explanations of supervised models that predict $\hat{S} = \hat{f}(P, F)$. In this example the training data for black-box models was generated by the ECM, but later we will see real data examples where this is not the case. Blue curves show how $\hat{S}$ depends on $P$ when $P$ is causally manipulated *without* holding $F$ constant, i.e. under the intervention $\text{do}(P = p)$. Orange curves show the dependence of $\hat{S}$ on $P$ when $F$ is held constant at its observed value, and coincide exactly with standard PDPs. Full definitions of these are given in Section 2. Several key takeaways:

- Comparing the direct (or partial) dependence curves and total dependence curves we see *there can be qualitative differences depending on the type of explanation desired*, for example one can be increasing while the other is decreasing. This is a consequential fact when considering how interventions may change (predicted) outcomes. Increasing $P$ causes larger values of $\hat{S}$, but if the increase in $P$ is done while holding $F$ constant then it could cause a decrease in $\hat{S}$ (or a smaller increase). **An intervention which does *not* hold other**

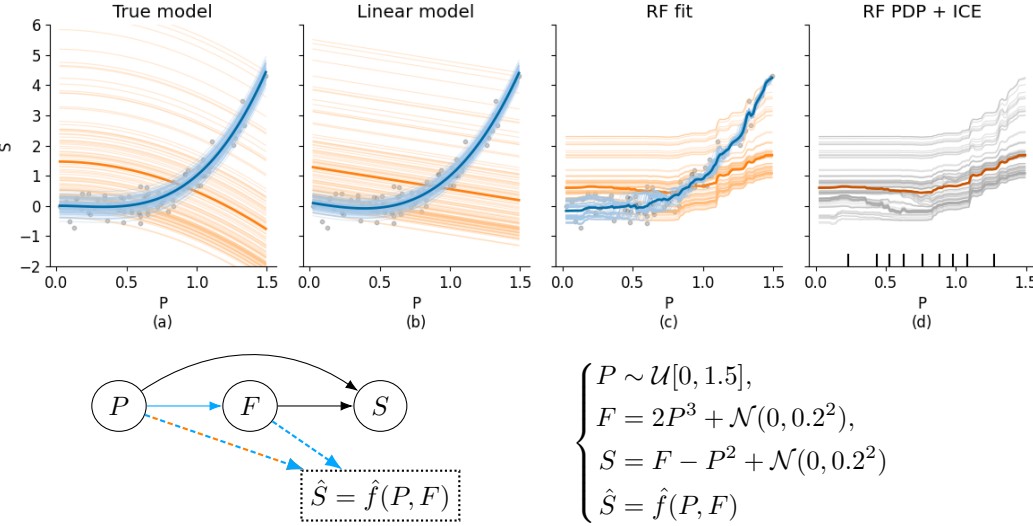

Figure 1: Motivating example. Causal Dependence Plots (top row) and the Explanatory Causal Model (bottom) for the motivating example. Points show the explanatory dataset, which in this example is also the training data for the predictive models. Counterfactual curves for individual points are shown as thin, light lines, with averages displayed as thick, dark lines. Total Dependence (TDP) is represented in blue and Natural Direct Dependence in orange. Panel (a) shows the relationships of the ECM. Panels (b-c) show CDPs for a linear model and random forest (RF) model, respectively. Panel (d) shows PDP and ICE curves for the RF model from a standard software library. This is identical to our NDDP in panel (c). We show this holds true in general: PDP/ICE are a special case of CDPs.

> **predictors constant—arguably the canonical causal operation—can be shown by our TDP**.
>
> - *Our framework includes some existing model explanation plots like ICE and PDPs as special cases*. In panels (c-d), and later in Theorem 2.10, we see that **PDP + ICE = NDDP**. A practitioner seeing only the PDP in panel (d) may conclude that "dependence" of $\hat{S}$ on $P$ is weak, especially if $P \leq 1$. The TDP in panel (c) shows a stronger increasing relationship closer to the true total dependence and a more holistic view of how $\hat{S}$ depends on $P$. Our work clarifies that the weaker form of dependence shown by PDPs is natural direct dependence. We also see the same weak dependence empirically for SHAP and ALE plots in Figure 4.
> - *Explanations of models can be qualitatively different from the underlying causal relationships*. For example, the random forest in panel (c) shows a direct dependence of $\hat{S}$ on $P$ that is increasing when the true direct dependence of $S$ on $P$ is decreasing. Predictive machine learning models may fail to capture causal dependence, and in this case studying the black-box would not necessarily help us learn about the real world. As another example, panel (b) shows that the total dependence of a linear model on a predictor can be non-linear, in this case because the mediator $F$ depends non-linearly on $P$.

## 1.1 Applications

Different combinations of the predictive setting and choice of ECM generate many uses for CDPs. In general, ECMs can be designed based on a particular desired explanation, make use of prior domain knowledge, or be learned and estimated from data using causal learning methods in a modular fashion. Importantly, the ECM does not need to contain the outcome variable $y$ except in one special case—residual plots—to be discussed later.

**Causal bridge.** In one special case we may wish to understand how $\hat{f}$ depends on a variable $z$ *which is not one of $\hat{f}$'s input features* but is causally related to them. Other methods cannot do this, but CDPs can provided the ECM also contains $z$. For simplicity we choose notation in the rest of the paper to reflect the case where the explanatory variable is an input, but this is not a loss of generality

since we can simply define $\hat{f}_{\text{drop}}(\mathbf{x}, z) := \hat{f}(\mathbf{x})$ and apply CDPs to $\hat{f}_{\text{drop}}$. Since the ECM may vary $\mathbf{x}$ when $z$ is changed, we can see how $\hat{f}$ depends on $z$. *This could be useful to probe a predictive model for fairness with respect to a sensitive attribute that the model does not use directly.*

**Incomplete causal knowledge.** There are various applications where an ECM does not need to be a fully specified or "correct" model for all features. First, CDPs only use the predicted—and not actual—outcome. This is useful for semi-supervised or anticausal learning: given causal structure *among predictors only* and a supervised learning model, attempt causal inference for the outcome [52, 63]. Second, we may only require explanations or plots for one or a small number of features. In such cases *we only need information from the ECM about interventions on the features of interest and their causal descendants*, and not other predictors. Finally, predictive models often use features that are known transformations or representations of inputs, and these transformations can be used to construct an ECM. For example, if the features are $(x, x^2, z)$, an ECM can simply encode the fact that $x^2$ depends causally on $x$. Even if we do not know the dependency between $x$ and $z$, we can make use of our partial knowledge about the features.

**Multiparty auditing, e.g. for fairness**. An owner of a predictive model may not have causal knowledge or incentives to use such knowledge. Predictive accuracy is their only concern. But a separate party, like a regulator, may audit that model. This party may have more causal knowledge due to specializing in auditing, or may be legally obliged to make certain causal assumptions for the purpose of the audit, e.g. to allow disparities only along certain causal pathways and not others. Previous work applied causality to fairness [5, 10, 27–29, 32, 34, 38, 48, 60, 62], recourse [7, 26, 44, 57], and other desiderata. *Existing methods like PDPs are limited to only showing direct dependence, and this may hide the full extent of unfairness or discrimination* [18]. CDPs can be used to probe a black-box for unfairness in the form of total dependence or partially controlled dependence.

**Explanations under covariate shift.** Often a pre-trained model is used for predictions on data from a different data generating process than the training DGP. We can use ECMs and CDPs to visualize how the model will behave out-of-distribution. ECMs could even be chosen adversarially.

**Scientific theory development.** Large and complex models may be fit to data where underlying structure is largely unknown. In such settings, relatively simple ECMs can be used to formulate simple hypotheses relating some predictors and plot causal dependencies to check these hypotheses or generate new ones. This can also be done hypothetically, assuming an ECM for exploration.

## 1.2 Contributions

After defining the CDP framework we demonstrate CDPs on synthetic and real datasets in Section 3 and Appendix B, including in conjunction with structural causal learning in B.2. We compare CDPs with other state of the art competitor visualization methods, for example in Figure 4. Theorem 2.10 establishes the first universally valid causal interpretation of PDP and ICE plots. Finally, in Section 2.7 we illustrate how to visualize uncertainty about the choice of ECM.

## 2 Methodology

### 2.1 Supervised learning models

We are given a predictive model $\hat{f}$, possibly estimated or learned using empirical risk minimization (ERM) $\hat{f} = \arg\min_{h \in \mathcal{H}} \sum_{i=1}^{n} \ell(h(\mathbf{x}_i), y_i)$ with some loss function $\ell$, pre-specified function class $\mathcal{H}$, and an independent and identically distributed training sample $\{(y_i, \mathbf{x}_i) : i = 1, \ldots, n\}$ with feature vectors $\mathbf{x}_i^T \in \mathbb{R}^p$. In Section 2.6 we focus on simple mediation analysis and partition the predictor variables into subsets so that $X$ and $M$ both notate predictors, $M$ being a mediator.

### 2.2 Fundamental problem of univariate explanations

To create an explanation of model dependence on a single feature, like a plot with $x_j$ on the horizontal axis and $\hat{f}$ on the vertical axis, *we must decide what to do with the other features* when varying $x_j$ along the plot axis. Most explanation methods use the same approach as the PDP and ICE plots: they *hold other features fixed* at values in a (auxiliary, explanatory) dataset. This may be unrealistic

if other features depend on $x_j$ causally, or even mathematically undefined if features are mutually constitutive, e.g. $(x, x^2)$ or the set $\{\mathrm{GDP}, \mathrm{GDP} \text{ per capita}, \text{population}\}$.

## 2.3 Structural Causal Models

Our notational conventions and definitions are influenced by [6, 41, 43]. Let $\mathbf{U}$ be a set of exogenous noise variables, $\mathbf{V}$ a set of $p = |\mathbf{V}|$ observable variables, and $\mathbf{G}$ a set of functions such that for each $j \in 1, \ldots, p$ we have $V_j = g_j(\mathbf{PA}_j, U_j)$, where $\mathbf{PA}_j \subseteq \mathbf{V}$ and $U_j \subseteq \mathbf{U}$ are the observable and exogenous parents, respectively, of variable $V_j$. Let the directed acyclic graph (DAG) $\mathcal{G}$ have vertices given by variables and, for each $V_j \in \mathbf{V}$ and each of the parent variables in $\mathbf{PA}_j$ and $U_j$, a directed edge oriented from the parents to $V_j$.

**Definition 2.1** (Structural Causal Model (SCM)). A (probabilistic) SCM $\mathcal{M}$ is a tuple $\langle \mathbf{U}, \mathbf{V}, \mathbf{G}, P_{\mathbf{U}} \rangle$ where $P_{\mathbf{U}}$ is the joint distribution of the exogenous variables. This distribution and the functions $\mathbf{G}$ determine the joint distribution $P^{\mathcal{M}}$ over all the variables in $\mathcal{M}$. Finally, causality in this model is represented by additional assumptions that $\mathcal{M}$ admits the modeling of interventions and/or counterfactuals as defined below.

**Definition 2.2** (Interventions). For the SCM $\mathcal{M}$, an intervention $I$ produces a modified SCM denoted $\mathcal{M}^{\mathrm{do}(I)}$ which may have different structural equations $\mathbf{G}^I$. Correspondingly, some variables may have different parent sets, so the DAG representation $\mathcal{G}^{\mathrm{do}(I)}$ may also change. We denote the new, interventional distribution as $P^{\mathcal{M};\mathrm{do}(I)}$. A simple class of interventions involves intervening on one variable, e.g.

$$I = \mathrm{do}\left(V_j := \tilde{g}(\tilde{\mathbf{PA}}_j, \tilde{U}_j)\right),$$

which changes how $V_j$ and all variables on directed paths from $V_j$ in $\mathcal{G}$ are generated. An even simpler sub-class of these are the atomic interventions setting one variable $V_j$ to one constant value $v$, which we denote $I_{j,v} := \mathrm{do}(V_j = v)$. Note that in this case $V_j$ has no parents in the graph $\mathcal{G}^{\mathrm{do}(I)}$; the source of the intervention itself is outside the world of the model.

Interventions are useful for modeling changes to a data generating process (DGP), for example, experiments that control a particular variable to see how its value changes other variables, or policy changes aimed at altering or removing existing causal relationships. In addition to generating new observations as a DGP, an SCM can also be used to model counterfactual values for observations that have already been determined. A counterfactual distribution is an interventional distribution defined over a specific dataset with information or constraints given by some of the observed values in that data, as we now describe.

**Definition 2.3** (Counterfactuals). Let $\mathbf{V}$ be the observed variables for observations in a given dataset, $\mathbf{PA}_j = \mathbf{v}$ and $U_j$ the observed and exogenous parents of variable $V_j$, and $I$ and intervention that modifies any of $V_j$'s parents. The intervention $I$ may hold some or all of $\mathbf{v}$ fixed and vary $U_j \leftarrow u$, passing these through $g_j(\mathbf{v}, u)$, or through $\tilde{g}_j(\tilde{\mathbf{v}}, u)$ if the intervention also changes any of $\mathbf{v} \leftarrow \tilde{\mathbf{v}}$. The counterfactuals $V_j(\tilde{\mathbf{v}}, u)$ are values $V_j$ would have taken if any of its observed and/or exogenous parents had taken the different values $(\tilde{\mathbf{v}}, u)$. To define the counterfactual distribution $P^{\mathcal{M}|\mathbf{V}=\mathbf{v};\mathrm{do}(I)}$, we use the posterior or conditional (depending on our probability model approach) distribution $P_{\mathbf{U}|\mathbf{V}=\mathbf{v}}$ to model uncertainty about $\mathbf{U}$ while computing counterfactual values of variables for an observation in the modified SCM $\mathcal{M}^{\mathrm{do}(I)}$.

*Remark* 2.4. Note that if the desired causal explanation uses counterfactuals, then we likely obtain the observed values from an auxiliary explanatory dataset. But since an SCM can generate data, we may also use it to generate the initial observed values and then re-use these when computing counterfactuals for the explanation.

## 2.4 Causal explanations

Our proposed solution to the fundamental problem highlighted for univariate explanations is to use an auxiliary ECM $\mathcal{M}_j$ and let this causal model determine how other features vary as functions of $x_j$. We denote these explanations as $\mathcal{E}_j(\hat{f}; \mathcal{M}_j)$ or $\mathcal{E}_j(\hat{f})$ if the context is clear. In the deterministic or noiseless case, suppose we know continuous functions $g_{kj}$ such that $x_k = g_{kj}(x_j)$, with $g_{jj}$ the identity. In this case the model $\mathcal{M}_j$ tells us $g(x_j) := (g_{1j}(x_j), \ldots, x_j, \ldots, g_{pj}(x_j))$ is a curve in $\mathbb{R}^p$ parameterized by $x_j$, and we generate the explanation $\mathcal{E}_j(\hat{f})$ by plotting $\hat{f}(g(x_j))$ against $x_j$.

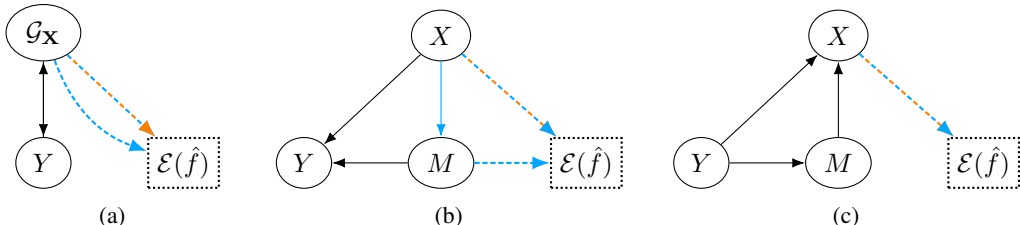

$$(a) \qquad\qquad\qquad (b) \qquad\qquad\qquad (c)$$

Figure 2: An ECM for predictors is used to produce an explanation $\mathcal{E}(\hat{f})$ of the predictive model $\hat{f}$. Solid arrows represent possible causal relationships in the ECM, and dotted arrows show dependence of the model explanation on predictors. In (a) $\mathcal{G}_{\mathbf{X}}$ denotes the subgraph of the SCM for predictors. In the mediation example (b), predictor $X$ causes $Y$ directly and also through mediator $M$, creating an important distinction between direct dependence (orange) and total dependence (blue). The reverse causality example (c) shows variables useful in predicting $Y$ may be caused by $Y$, and also be causes of prediction $\hat{Y} = \hat{f}(X, M)$ and the explanation of that prediction.

To extend this strategy to non-deterministic causal models we use an ECM $\mathcal{M}_{\mathbf{X}}$ for the predictor variables with $\mathcal{G}_{\mathbf{X}}$ its associated DAG. We represent this graphically in Figure 2. The expressive power of SCMs allows us to pose various interpretive questions and compute various kinds of explanations by performing operations in $\mathcal{M}_{\mathbf{X}}$.

## 2.5 Causal Dependence Plots

For the following definitions, we assume predictor variables $\mathbf{X} \in \Omega_{\mathbf{X}}$, an outcome of interest $Y \in \Omega_Y$, and a black-box function $\hat{f}(x) : \Omega_{\mathbf{X}} \to \Omega_Y$ with outputs that we may also denote $\hat{Y}$. A structural causal model $\mathcal{M}$, either assumed or learned from data, specifies causal relationships for the predictors $\mathbf{X}$, i.e. it need not involve the outcome $Y$. Note that predictors may only be a subset of the variables in $\mathcal{M}$ as in the "causal bridge" application discussed in Section 1.1.

**Definition 2.5** (Explanatory Causal Model (ECM)). An ECM $\mathcal{M}'$ augments the SCM containing predictors by including the predicted outcome $\hat{Y}$ as an additional variable with $\hat{f}$ as its structural equation.

Generating causal explanations for $\hat{f}$ involves performing abduction, action, and prediction with this ECM. In a large ECM graph we may suppress all arrows into $\hat{Y}$ except those from the explanatory feature and its descendants. This is to simplify the display, as in Figure 5.

**Additional notation and conventions.** We use the shorthand $\hat{f}(P^{\mathcal{M}})$, where $\hat{f}$ takes a distribution $P^{\mathcal{M}}$ as its argument, to denote using data from that distribution as the input to the black-box function $\hat{f}$. For each type of causal explanation with a given *Named Effect* based on intervention $I$, we define the *Individual Counterfactual Named Effect* curves as the set of counterfactual curves $\hat{f}(P^{\mathcal{M}|\mathbf{V}=\mathbf{v};\mathrm{do}(I)})$ for each individual, the *Named Effect Function* as their (empirical) expectation $\hat{\mathbb{E}}\left[\hat{f}(P^{\mathcal{M}|\mathbf{V}=\mathbf{v};\mathrm{do}(I)})\right]$, and the *Named Dependence Plot* as a plot displaying all of these curves.

**Definition 2.6** (Causal Dependence Plot (CDP)). Given a function $\hat{f}$, explanatory dataset $\mathcal{D}$, ECM $\mathcal{M}$, and family of interventions $I_\theta$ parameterized by $\theta$, we construct a plot with $\theta$ as the horizontal axis and display Individual Counterfactual (IC) curves

$$\mathsf{IC}(\theta) = \hat{f}(P^{\mathcal{M}|\mathbf{X}=\mathbf{x};\mathrm{do}(I_\theta)}). \tag{2}$$

These show the effect of intervention $I_\theta$ on black-box output for each individual observation in the explanatory dataset as $\theta$ varies. The (empirical) average of these (over the explanatory data) is (an estimate of) the Causal Effect Function (CEF), and a plot showing the IC and CEF is a Causal Dependence Plot.

We typically apply this to create plots for one explanatory feature $X_s$ at a time using interventions like $I_\theta = \mathrm{do}(X_s = \theta)$. Horizontal axes for plots use a grid over the possible values of $X_s$ given by its range in dataset $\mathcal{D}$. Bar graphs can be used when the explanatory feature is categorical.

Next is perhaps the most straightforward and important named effect.

**Definition 2.7** (Total Dependence Plot (TDP)). For an intervention $I$, the Individual Counterfactual Total Effect (ICTE) curves

$$\mathsf{TE}(I) = \hat{f}(P^{\mathcal{M}|\mathbf{X}=\mathbf{x};\mathrm{do}(I)}) \tag{3}$$

show the total effect of intervention $I$ on black-box output for each individual observation in the explanatory dataset. The (empirical) average of these (over the explanatory data) is (an estimate of) the Total Effect Function (TEF), and a plot showing the ICTE and TEF is a Total Dependence Plot (TDP). We compute the TDP following Algorithm 1.

---

**Algorithm 1** Total Dependence Plot (TDP)

Inputs: $\mathcal{M}$ (ECM), $\hat{f}$ (black-box predictor), $\mathcal{D}$ (explanatory dataset), $X_s$ (covariate of interest)

> Let $X$ be a grid of possible values of $X_s$
> Set $N$ to the number of observations in $\mathcal{D}$
> Initialize $N \times |X|$ matrix of predictions $\hat{Y}$
> **for** $x$ in $X$ **do**
>      Define intervention $I = \mathrm{do}(X_s = x)$
>      Sample counterfactual dataset $\mathcal{D}_{X_s \leftarrow x}$ entailed by $P^{\mathcal{M}|D;\mathrm{do}(I)}$
>      Set $\hat{Y}[:, x]$ to $\hat{f}(\mathcal{D}_{X_s \leftarrow x})$
> **end for**
> Plot $N$ lines $(X, \hat{Y}[i, :])$ {(Individual Counterfactuals)}
> Plot average $(X, \sum_i \hat{Y}[i, :]/N)$ {(Causal Dependence)}

---

*Remark* 2.8. In the remaining definitions, we give notation only for the individual counterfactual curves and leave the other objects implicitly defined.

We often wish to decompose how much of the total effect of $X$ on $\hat{Y}$ is attributable to different pathways between the variables. This can be explored via direct dependence below, as well as with other named CDPs described in Appendix A.

**Definition 2.9** (Natural Direct Dependence Plot (NDDP)). Given intervention $I$ define a corresponding intervention $J$ that intervenes on all descendants of any variables that are changed by $I$, except for $\hat{Y}$, and resets them to their observed values in dataset $\mathcal{D}$. We then define the Individual Counterfactual Natural Direct Effect curves

$$\mathsf{NDE}(I) = \hat{f}(P^{\mathcal{M}|\mathbf{X}=\mathbf{x};\mathrm{do}(I,J)}). \tag{4}$$

This quantity represents the effect of intervention $I$ on black-box output $\hat{Y}$ while all variables not directly intervened upon by $I$ are fixed at their 'natural,' i.e., pre-intervention values in $\mathcal{D}$. Algorithm 4 demonstrates how to compute the NDDP.

From this construction of NDDP, we see by comparing it to the **PDP and ICE algorithm** that it is equivalent to these, confirming what we observed in Figure 1(d).

**Theorem 2.10** (PDP + ICE = NDDP). *When generating plots for the predictive model $\hat{f}$ using the dataset $\mathcal{D}$ and feature $X_s$, the ICE plot curves and Individual Counterfactual Natural Direct Dependence curves are identical. Hence, the NDDP is identical to a PDP that includes ICE curves.*

*Proof.* We have implicitly assumed both plots will use the same range for their horizontal axes. This is natural as implementations use the range of the feature in the dataset, and both plots are constructed from the same feature $X_s$ in the same dataset $\mathcal{D}$. Since the PDP and NDDP both contain empirical averages of their respective Individual curves it suffices to show these are equal at each point $\tilde{x}$ in the plot grid.

Consider individual $i$ in dataset $\mathcal{D}$ with features $\mathbf{x}_i = (x_{i1}, \ldots, x_{ip})$. The value of the ICE curve at $\tilde{x}$ for this individual is, from (1), $\hat{f}(\mathbf{x}'_{i,k})$ where $\mathbf{x}'_{i,k} := (x_{i1}, \ldots, x_{is} \leftarrow \tilde{x}, \ldots, x_{ip})$, i.e. the original features $\mathbf{x}_i$ but with entry $s$ set to $\tilde{x}$. We must show this is the same as the value of the Individual Counterfactual Natural Direct Dependence curve at $\tilde{x}$ for individual $i$. Applying Definition 2.9, we use interventions

$$I = \mathrm{do}(X_s = \tilde{x}) \text{ and } J = \mathrm{do}(X_j = \mathbf{x}_j \text{ if } X_s \text{ is an ancestor of } X_j).$$

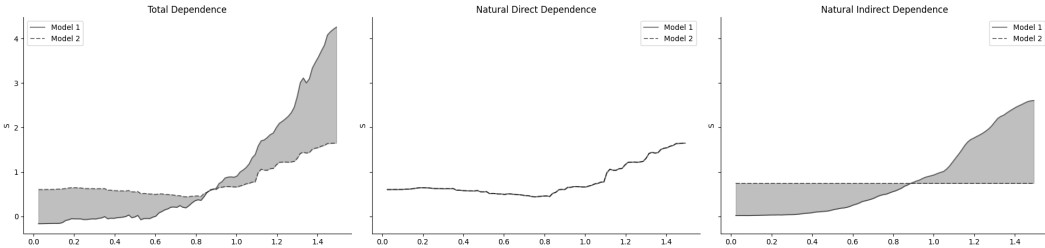

Figure 3: TDP, NDDP, and NIDP uncertainty bands for the salary example using the random forest model in Figure 1. The range of curves is induced by two candidate ECMs described in Section 2.7.

The NDDP applies these in the order $I$ followed by $J$. First, $I$ sets the value of feature $X_s$ to $\tilde{x}$ for all individuals, and may modify other features if they are descendants of $X_s$. Then $J$ intervenes on each descendant $X_j$ of $X_s$ and resets it to its observed values in $\mathcal{D}$, and in particular for individual $i$ these are each reset to $x_{ij}$. Hence, the value of the Individual Counterfactual Natural Direct Dependence curve at $\tilde{x}$ for individual $i$ is also given by $\hat{f}(\mathbf{x}'_{i,k})$. $\qquad\square$

*Remark* 2.11. Note that there is some subtlety in the assumption of using the same dataset: a Bayesian probability modeling approach to SCMs may add more randomness when sampling counterfactuals. In this case, rather than the ICE and ICNDD curves being identical, the ICE will equal the expectation of the ICNDD curves over this additional source of randomness. The additional randomness is specified by priors over exogenous variables, and the expectation can be estimated by more sampling at a computational cost of a constant factor. Since the usage of these plots is visual and somewhat qualitative this subtlety is not an important limitation of the theorem, and it would only apply under particular modeling assumptions.

*Remark* 2.12. To our knowledge this is the first result establishing a universally valid causal interpretation of PDPs. **Its most important limitation is that it applies to the model output $\hat{Y}$ and not necessarily the original outcome $Y$**.

Several other types of named CDPs are described in Appendix A.

## 2.6 Mediation analysis

Many applications involve a causal structure we refer to as a mediation triangle, with examples shown in Figure 1 and Figure 2b. In mediation analysis, we often wish to decompose how much of the total effect of $X$ on $Y$ is attributable to the pathway through $M$ and how much of it is direct. CDPs allow us to visualize frequently studied quantities of interest in this setting including other special cases defined in Appendix A.1. Although mediation analysis motivates CDPs and helps build intuition, we emphasize that our definitions can be used *with any structural causal model*. See Section 3 for other, more complex examples.

## 2.7 Uncertainty and sensitivity analysis

There are various ways to incorporate uncertainty about the ECM into CDPs. We explore a natural first extension of the CDP that shows a *range between possible effect functions* induced by a *set of auxiliary ECMs*. The set of ECMs could be pre-specified or, for example, given by a Markov equivalence class output by a causal structure learning algorithm. Returning to our motivating example from Section 1, we might question whether parental income $P$ impacts school funding $F$, considering instead an SCM without mediation: $P \to S \leftarrow F$. Figure 3 shows a range of possible effect functions interpolating between this ECM without the indirect effect and the original ECM in Section 1, for each of the TDP, NDDP, and NIDP. In this we have assumed the same structural equations for the edges that are common to both models. These plots show a range for how $\hat{S}$ may depend on $P$ when we are unsure how $F$ depends on $P$. Figure 9 in Appendix B.2 shows an example with real data where we use candidate ECMs discovered by the PC algorithm. These examples are not confidence regions, but any method for producing confidence sets in SCMs could also be used with CDPs to display uncertainty regions. Future work can develop additional methods for

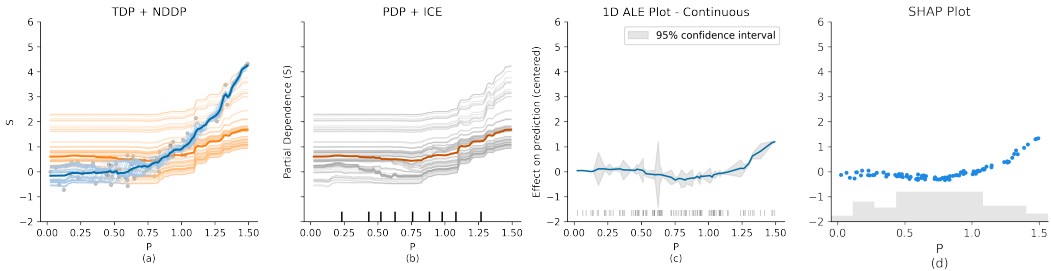

Figure 4: Comparison of CDP (a) with PDP (b), ALE (c), and SHAP plots (d) for the salary example in Figure 1. Our TDP stands out, and all other plots are qualitatively similar.

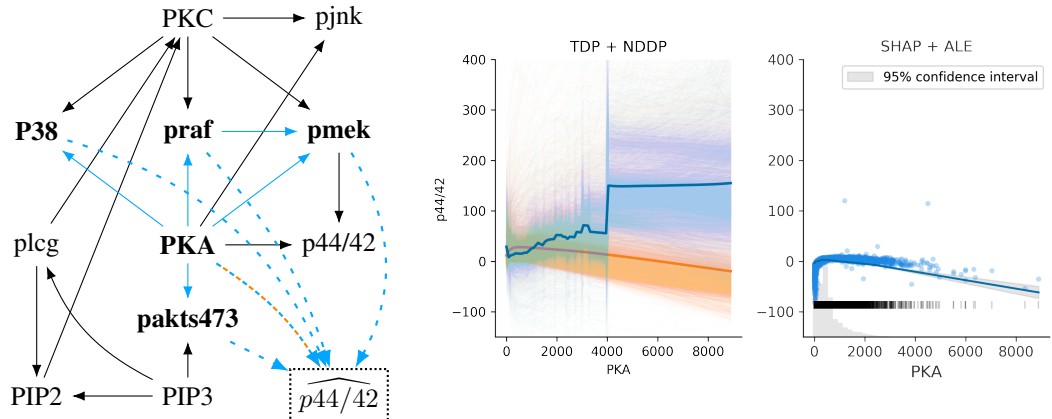

Figure 5: ECM for the Sachs et al. [49] dataset (left), CDPs for an MLP predictive model (center), ALE (line) and SHAP (points) plots (right). All plots visualize the effect of PKA on predicted p44/42. PKA and its descendants are bolded. The NDDP (i.e. PDP + ICE), ALE, and SHAP all show an overall decrease, while the TDP shows an increase. *Conclusions depend strongly, qualitatively, on the specific interpretive question we ask, and causal modeling allows us to formulate questions precisely.*

visualizing uncertainty, for example by leveraging sensitivity analysis based on conformal prediction [9, 21, 31, 61].

## 3 Experiments

We demonstrate CDPs in a series of experiments with simulated and real datasets.

**Comparison with other explanatory plots.** Figure 4 shows accumulated local effect (ALE) [3] and Shapley Additive Explanation (SHAP) [33] feature plots for the salary example. These appear similar to the PDP, with a more weakly increasing relationship than that seen in the TDP. TDPs represent a significant and novel contribution to the existing model visualizations.

**Real data with domain knowledge.** An ECM may be constructed using domain expertise. Figure 5 shows an ECM and CDPs for the Sachs et al. [49] dataset of expression levels of proteins and phospholipds in human cells, for which data and a ground-truth DAG[1] are publicly available in the Causal Discovery Toolbox [24]. While the actual biology of the problem is not our focus here, there are meaningful takeaways from the figure. For this model, the TDP shows an increasing relationship, while the NDDP/PDP shows a decrease. *The overall direction of the trend in predictions based on PKA is reversed if we hold other predictors fixed.* This is an important lesson for using model explanations in scientific machine learning.

---

[1]Following the discussion in [45] and follow-up ground truth DAG for the Sachs et al. [49] dataset in Figure 5 of [45], we choose the edge PIP3 → PIP2 in order to eliminate a would-be cycle.

Additional experiments can be found in Appendix B. Notably, in Appendix B.2 we *learn an ECM from data with a causal structural learning algorithm* and then use it to produce CDPs. The main takeaway of the real data experiments is that CDPs can be useful in practice.

In simulation experiments we know the true DGP, so we can compare its functional form to various black-box models and their explanatory plots. Results in Appendix B.1 show that CDPs are sensitive to whether the functional form assumptions of the black-box model fit the DGP. In other words, *if a black-box model is a poor fit to the DGP, then CDPs can accurately explain the black-box but will not reflect the true DGP*. This limitation is not specific to CDPs but applies to all explanation methods. Figure 6 also shows that different ECMs can produce different CDPs for the same black-box model, and *a misspecified ECM can produce misleading CDPs*.

## 4 Discussion

**Related work.** Recent work in recourse [25] uses contrastive or counterfactual explanations [56]. Some of this focuses on causal dependence [50]. Blöbaum and Shimizu [4] identify the predictor with the largest total effect, which is most applicable when assuming linearity. Zhao and Hastie [63] investigated causal interpretations of PDP, aiming for causal inference about the underlying DGP, and showed that when the DGP satisfies the backdoor criterion [39] then a PDP visualizes the total effect (TE) of a predictor. Cox Jr [11] observed an association between partial dependence plots and NDE, an equivalence we formally establish in Theorem 2.10, to our knowledge the first such result. Lazzari et al. [30] weight observations when computing PDPs. There has been some recent work creating causal variants of SHAP [1, 13, 19, 23, 58], and in future work we will explore comparisons of appropriate special cases of CDPs with these. We are not aware of any previous causal explanation work with the generality of CDPs.

**Limitations.** Causal modeling always involves some limitations [15, 47]. For CDPs, full specification of an ECM can be a strong assumption. However, in Sections 1.1 and 2.7 we discussed some ways this can be relaxed. In general, *if a causal explanation is desired or necessary, then we cannot avoid making causal assumptions*. Model-agnostic explanation methods also always have certain limitations [2, 36]. For example, if the predictive model fails to fit the DGP, then any model explanation will also fail if our interpretive goal is to learn about the DGP [63]. *CDPs may be misleading if the true DGP differs in important ways from the ECM*, as shown in Figure 6. However, standard PDPs and similar explanation methods also require auxiliary explanatory data, and that data may also differ from the target DGP. So this is not an additional limitation specific to our method.

**Conclusion.** Causal Dependence Plots use an explanatory causal model to create plots with causal interpretations. This allows us to use the powerful language of structural causal models to pose and answer a variety of meaningful questions. Our framework generalizes Partial Dependence Plots, which Theorem 2.10 shows it includes as a special case, and allows other kinds of causal interpretations we have not seen previously explored in the literature. Future work in this direction could expand on some canonical causal structures for useful applications, or interface with other kinds of models, for example extending to non-tabular data by applying causal representation learning. Relating explanation methods to Pearl's ladder of causation [40], most previous interpretable machine learning and explainable AI methods—like PDPs—concern associations and hence are confined to the first rung of the ladder. With CDPs we ascend the ladder, creating model interpretations intended to change the world. Interpretability provided the initial motivation for CDPs, but since plots are qualitative CDPs also open the door for future work on causal methodology that relaxes assumptions while maintaining *visual validity*.

**Broader Impacts.** There are many potential societal consequences of our work: essentially those shared by all tools for model interpretability and explainability. Model explanations can be misleading, either due to error or intentional deception. When a user is convinced by a flawed model explanation to reach misguided conclusions about a model, they may make harmful or sub-optimal decisions about how or whether to use that model. For example, if an explanation tool is used to assess the fairness of a model, a flawed explanation could lead to the conclusion that a discriminatory model is fair or that a fair model is discriminatory. In applications related to science, a flawed model explanation can lead to wasting resources pursuing a dead-end hypothesis or to missing out on an important discovery. Similarly, poor business decisions can be made on the basis of flawed explanations.

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

# A Methodology supplement

## A.1 Additional CDPs

For completeness we include here definitions for several other important, named CDPs.

**Definition A.1** (Partially Controlled Dependence Plot (PCDP)). Consider intervention $I$ affecting a subset of variables in $\mathcal{G}_{\mathbf{X}}$ and atomic intervention $C$ that holds constant a disjoint subset of variables. The Individual Counterfactual Partially Controlled Effect curves

$$\mathsf{PCE}(I, C) = \hat{f}(P^{\mathcal{M}|\mathbf{X}=\mathbf{x};\mathrm{do}(I,C)}) \tag{5}$$

represent the effect of intervention $I$ on black-box output $\hat{Y}$ while other variables are set (via intervention) to specific constant values. We compute the PCDP via Algorithm 3.

**Definition A.2** (Natural Indirect Dependence Plot (NIDP)). Consider atomic intervention $I$ and a corresponding intervention $K$ that removes from $\mathcal{G}_{\mathbf{X}}$ all outgoing edges from any of the nodes intervened upon by intervention $I$ and sets those nodes to their observed values in the explanatory dataset. For example, if $I = \mathrm{do}(A = a, B = b)$, then intervention $K$ will remove all outgoing edges from $A$ and $B$ and set $A$ and $B$ to their original observed values. We then define Individual Counterfactual Natural Indirect Effect curves

$$\mathsf{NIE}(I) = \hat{f}(P^{\mathcal{M}_{\mathbf{x}}^{\mathrm{do}(I)}|\mathbf{X}=\mathbf{x};\mathrm{do}(K)}). \tag{6}$$

Notice that intervention $I$ is performed before intervention $K$. This quantity represents the effect of intervention $I$ on black-box output $\hat{Y}$ that is due only to any indirect pathways to $\hat{Y}$. We compute the NIDP following Algorithm 5. The difference between two values of this function can be used to express the natural indirect effect as a special case.

## A.2 Algorithms

This section describes additional algorithms including several special cases of named CDPs.

---

**Algorithm 2** Explanatory Causal Model (ECM)

Inputs: $\mathcal{M}$ (SCM), $\hat{f}$ (black-box predictor), $\mathbf{S} \subseteq \mathbf{X}$ (covariates used by black-box)

Output: $\mathcal{M}'$ (SCM)

---

    Make copy $\mathcal{M}'$ of SCM $\mathcal{M}$ and perform all subsequent operations on this copy

    Add node for $\hat{Y}$ to causal graph $\mathcal{G}'$ of SCM $\mathcal{M}'$

    **for** $x$ in $\mathbf{S}$ **do**

        Add edge in $\mathcal{G}'$ from $x$ to $\hat{Y}$

    **end for**

    Set structural equation for node $\hat{Y}$ to $\hat{f}$

    Set exogenous variable $U_{\hat{Y}}$ to 0

---

**Algorithm 3** Partially Controlled Dependence Plot (PCDP)

Inputs: $\mathcal{M}$ (ECM), $\hat{f}$ (black-box predictor), $\mathcal{D}$ (explanatory dataset), $X_s$ (covariate of interest), $C$ (intervention controlling other variables in $\mathcal{M}$)

---

    Let $X$ be a grid of possible values of $X_s$

    Set $N$ to the number of observations in $\mathcal{D}$

    Initialize $N \times |X|$ matrix of predictions $\hat{Y}$

    **for** $x$ in $X$ **do**

        Define intervention $I = \mathrm{do}(X_s = x, C)$

        Sample counterfactual dataset $\mathcal{D}_{s \leftarrow x, C}$ entailed by $P^{\mathcal{M}|D;\mathrm{do}(I)}$

        Set $\hat{Y}[:, x]$ to $\hat{f}(D_{s \leftarrow x, C})$

    **end for**

    Plot $N$ lines $(X, \hat{Y}[i, :])$ {(Individual Counterfactuals)}

    Plot average $(X, \sum_i \hat{Y}[i, :]/N)$ {(Causal Dependence)}

---

**Algorithm 4** Natural Direct Dependence Plot (NDDP)

Inputs: $\mathcal{M}$ (ECM), $\hat{f}$ (black-box predictor), $\mathcal{D}$ (explanatory dataset), $X_s$ (covariate of interest)

---

Let $X$ be a grid of possible values of $X_s$
Set $N$ to the number of observations in $\mathcal{D}$
Initialize $N \times |X|$ matrix of predictions $\hat{Y}$
Get all descendants of $X_s$ in $\mathcal{M}$, excluding $\hat{Y}$, and store in $\mathbf{C}$
Get observed values of all variables in $\mathbf{C}$ and store in $\mathbf{c}$
Define intervention $J = \text{do}(\mathbf{C} = \mathbf{c})$
**for** $x$ in $X$ **do**
    Define intervention $I = \text{do}(X_s = x)$
    Sample counterfactual dataset $\mathcal{D}_{X_s \leftarrow x}$ entailed by $P^{\mathcal{M}|D;\text{do}(I,J)}$
    Set $\hat{Y}[:,x]$ to $\hat{f}(\mathcal{D}_{X_s \leftarrow x})$
**end for**
Plot $N$ lines $(X, \hat{Y}[i,:])$ {(Individual Counterfactuals)}
Plot average $(X, \sum_i \hat{Y}[i,:]/N)$ {(Causal Dependence)}

---

**Algorithm 5** Natural Indirect Dependence Plot (NIDP)

Inputs: $\mathcal{M}$ (ECM), $\hat{f}$ (black-box predictor), $\mathcal{D}$ (explanatory dataset), $X_s$ (covariate of interest)

---

Let $X$ be a grid of possible values of $X_s$
Set $N$ to the number of observations in $\mathcal{D}$
Initialize $N \times |X|$ matrix of predictions $\hat{Y}$
Get all descendants of $X_s$ in $\mathcal{M}$, excluding $\hat{Y}$, and store in $\mathbf{C}$
Make copy $\mathcal{M}'$ of SCM $\mathcal{M}$
**for** $x$ in $\mathbf{C}$ **do**
    Remove all incoming edges to $x$ from $\mathcal{M}'$
**end for**
Define intervention $I = \text{do}(X_s = x)$
**for** $x$ in $X$ **do**
    **for** $i$ in $N$ **do**
        Sample counterfactual observation $d_c$ for unit $i$ entailed by $P^{\mathcal{M}|D[i];\text{do}(I)}$
        Get counterfactual values of all variables in $\mathbf{C}$ from observation $d_c$ and store in $\mathbf{c}_i$
        Define intervention $J = \text{do}(X_s = x, \mathbf{C} = \mathbf{c}_i)$
        Sample counterfactual observation $d_c'$ for unit $i$ entailed by $P^{\mathcal{M}'|D[i];\text{do}(J)}$
        Set $\hat{Y}[i,x] \leftarrow d_c'[y]$ for index $y$ corresponding to node $\hat{Y}$
    **end for**
**end for**
Plot $N$ lines $(X, \hat{Y}[i,:])$ {(Individual Counterfactuals)}
Plot average $(X, \sum_i \hat{Y}[i,:]/N)$ {(Causal Dependence)}

---

# B Experiments supplement

## B.1 Model misspecification

Consider the non-linear mediation example with the following DGP:

$$
\begin{aligned}
U_X, U_{M_1}, U_{M_2}, U_Y &\sim \mathcal{N}(0,1) \\
X &= U_X \\
M_1 &= \frac{1}{2}X^3 + U_{M_1} \\
M_2 &= \frac{1}{4}X^3 + U_{M_2} \\
Y &= M_1^2 + M_2^2 - \frac{1}{2}X^2 + U_Y.
\end{aligned}
\tag{7}
$$

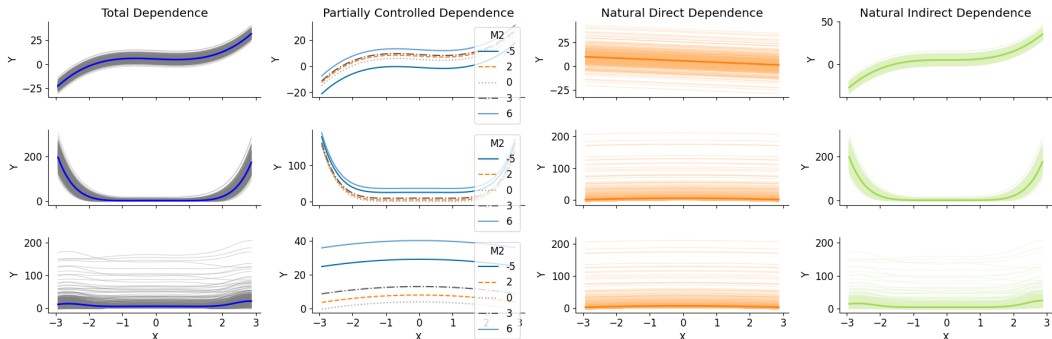

Figure 6: CDPs for the simulation example with data from (7), shown for a 'good' black-box and correct ECM (top row), a 'bad' black-box model and correct ECM (middle row), and a 'good' black-box and misspecified ECM (bottom row).

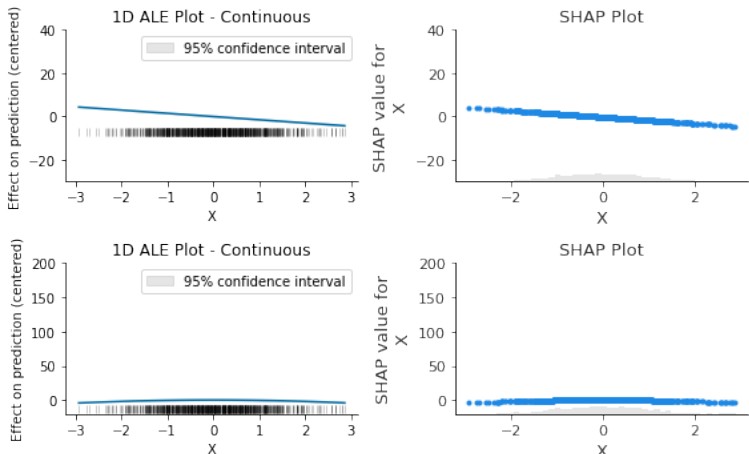

Figure 7: ALE and SHAP for the simulation example with data from (7), shown for a 'bad' black-box model (top row), and a 'good' black-box (bottom row). Both ALE and SHAP show a similar relationship as NDDP for both models.

We use this DGP to fit two different black-box models: one model that assumes the correct functional form (i.e., the relationship for $Y$ shown in the DGP above), and an 'incorrect' model that predicts $Y$ via linear regression. We use two different ECMs to construct CDPs, one which is the true DGP and one which incorrectly assumes the structure $M_1 \rightarrow X \rightarrow M_2$. Figure 6 shows the CDPs for each of these models using the black-box training data as the explanatory data. Similarly, Figure 7 shows SHAP and ALE plots using the same explanatory data.

We can glean a couple insights from Figure 6. First, CDPs are sensitive to whether the functional form assumptions of the black-box model fit the ground truth data generating process. Good explanations for $\hat{Y}$ may be different from good explanations for $Y$ if the black-box model is poorly specified. The second is that a misspecified ECM can produce bad explanations even when the black-box model correctly fits the true causal relationships in the DGP.

In Figure 7 we see that ALE and SHAP produce similar explanations to the PDP/NDDP. We saw this previously for the random forest black-box model in the example from Figure 1.

## B.2 Real data with structural causal learning

The Breast Cancer Wisconsin (Original) dataset [35] is a publicly available dataset often used to test algorithms on medical data. The dataset contains 9 ordinal variables, which represent attributes of the cells within a breast mass: Clump Thickness, Uniformity of Cell Size, Uniformity of Cell Shape,

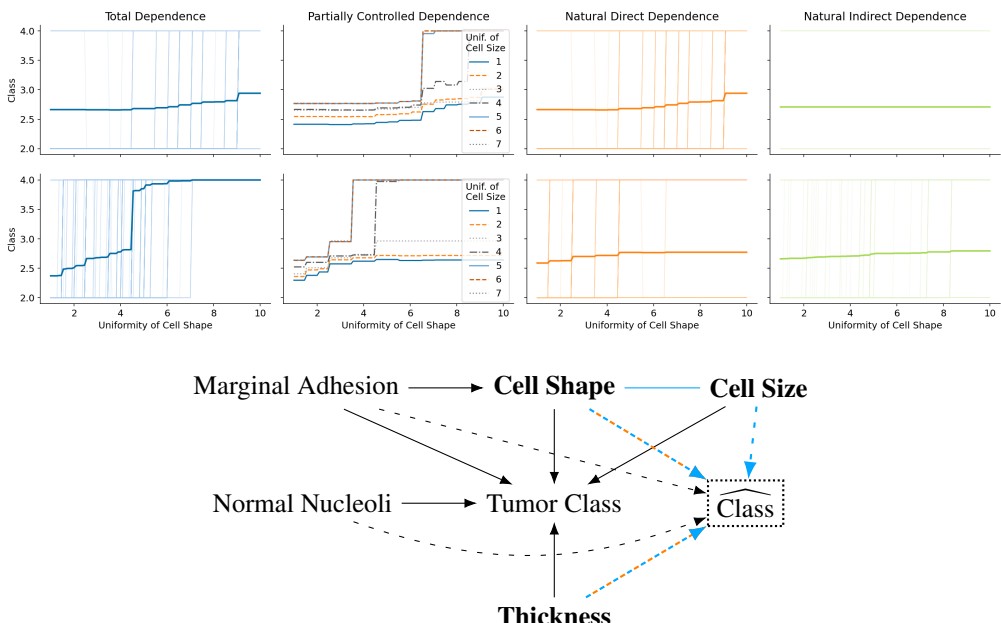

Figure 8: Breast cancer data example. CDPs for a random forest classifier and predictors Clump Thickness (first row) and Uniformity of Cell Shape (second row). Structural graph $\mathcal{G}_B$ for the ECM learned by the PC algorithm (last row). The outcome Class is binary: 2 for benign, 4 for malignant. For the undirected edge between Cell Size and Cell Shape, we investigate the sensitivity to the different options in Figure 9. Note: Our intention is not to make conclusive scientific statements, but only to demonstrate how CDPs could be used in conjunction with causal structure learning.

Marginal Adhesion, Single Epithelial Cell Size, Bare Nuclei, Bland Chromatin, Normal Nucleoli and Mitoses. The outcome variable is the class of the breast tumor, benign or malignant.

We use a causal structural learning algorithm, specifically the PC algorithm [55] implemented in Julia `CausalInference` [51], to learn a DAG for this dataset, on a smaller subset of predictor variables for simplicity. Figure 8 shows the resulting DAG and CDPs for a random forest model to classify the Class variable.

This shows CDPs can be combined with other causal methods like structural learning algorithms. The PC algorithm output had an undirected edge between Cell Size and Cell Shape. We next explore the graph structures consistent with this uncertain edge.

Figure 9 shows the TDP, NDDP, and NIDP for a learned additive noise model with three different structures consistent with $\mathcal{G}_B$: (1) with the edge Cell Shape $\rightarrow$ Cell Size, (2) with the edge Cell Size $\rightarrow$ Cell Shape, and (3) with no edge between Cell Size and Cell Shape. This figure shows that the takeaway about cell shape impacting tumor class is indeed sensitive to our choice about the uncertain edge, particularly for the TDP.

### B.3 Individual curves showing heterogeneity

One criticism of PDPs is that they may hide heterogeneity or individual variation, and for this reason it may be good practice to include the ICE curves in any PDP. Just like the individual curves in an ICE plot, the individual counterfactual curves in our CDPs can show important effects that are hidden by averaging as illustrated in Figure 10. In our implementation the default settings for CDPs—and our recommendation—is to show these individual curves.

### B.4 Causal dependence for residuals

CDPs are applicable not only to understand model outputs but also to understand model performance. For example, CDPs can be used to probe residuals (or other measures of error) under distribution

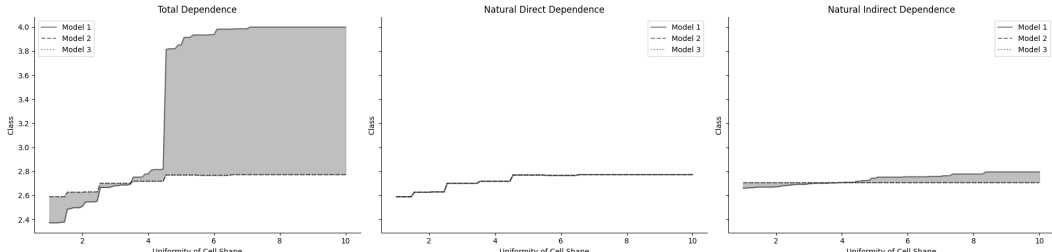

Figure 9: Total Dependence Plots, Natural Direct Dependence Plots and Natural Indirect Dependence Plots for the Breast Cancer Wisconsin dataset under three possible DAGs consistent with the PC algorithm output: (1) with the edge Cell Shape $\to$ Cell Size, (2) with the edge Cell Size $\to$ Cell Shape, and (3) with no edge between Cell Size and Cell Shape.

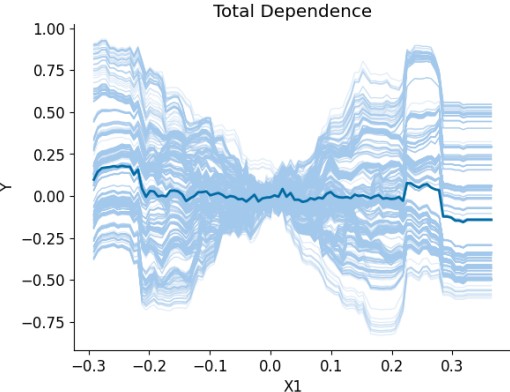

Figure 10: Individual counterfactual curves can show heterogeneous effects. In this example the relationship is positive for some individuals and negative for others with average effect of zero.

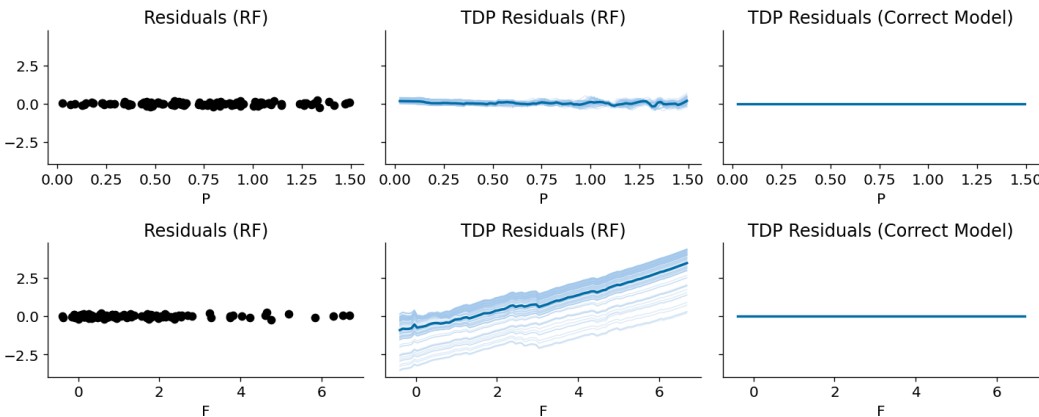

Figure 11: Regular versus CDP residuals for the example in Figure 1, plotted against feature $P$ in the top row and $F$ in the bottom row. Model multiplicity means two models can produce nearly the same predictions, with high accuracy, while using different functional relationships. Accuracy can only show if the model is "observationally correct," (left column) while CDPs can help determine if the model is also "causally correct" (middle vs right columns)
.

shift. Figure 11 demonstrates this with the salary example in Figure 1. Using the same RF model, residuals show no trend with respect to parental income $P$ nor school funding $F$. However, a TDP over the residuals reveals the random forest model learns a causally incorrect functional form for $\hat{S}$, even if it fits the training data well. By comparison, only the causally correct model shows no trend in residuals with a CDP. This is an empirical verification of the correspondence between robustness to distribution shift and causal learning [46].

## C   Code and reproducibility

Predictive models were fit using `scikit-learn` [42]. CDP implementations make use of causal modeling functions in `dowhy` [53]. Figures were generated with `matplotlib` [20]. For causal structural learning we used the PC algorithm [55] implemented in Julia `CausalInference` [51]. We used the Python implementation of ALE plots in [22]. In experiments we used the Breast Cancer Wisconsin (Original) dataset [35, 59] and Sachs et al. [49] dataset. No specialized hardware is required to run the experiments as they are not computationally costly and can be reproduced on a personal computer. Our code to implement CDPs, run the experiments, and produce figures is available at this repository: `https://github.com/causalhypothesis/cdp-neurips/`

