# OpenReview forum: "Causal Dependence Plots"
_NeurIPS.cc/2024/Conference — NeurIPS 2024 poster_

### Official Review · Reviewer_GtVe · 2024-06-21

**Soundness:** 3
**Presentation:** 3
**Contribution:** 2
**Rating:** 6
**Confidence:** 3

**Summary:**

The paper introduces causal dependence plots to visualize how (black-box) model predictions are affected by changes in input variable distributions. The explanations are generated based on an underlying explanatory causal model and can be generated for different types of causal quantities such as direct or indirect effects.

**Strengths:**

1. The paper introduces a new explanatory plotting feature, which leverages knowledge about the underlying causal model to identify different types of causal quantities. The method builds upon known results in the causal inference literature and nicely embeds these quantities in a general explanatory framework.

2. All claims are well supported and the paper is mostly well structured. The visualizations make it easy to follow the ideas and results

3. Leveraging causal structures for explanatory plots is a relevant topic and enables detailed insights and model explanations.

**Weaknesses:**

1. The authors mention the limitation of a fully specified ECM. As this not only requires complete knowledge of the DAG but also the functional form of the corresponding functions G, this is a strong assumption. I think the paper would benefit, if the authors discuss the case if these functions are estimated instead. This severely reduces applicability and clarity of the approach.

Minor Comments:
- L. 45: Define $\hat{f}$ before
- L. 46: typo in “containsing”
- Please define abbreviations such as TDP (l. 68) and NDDP (l. 71) before mentioning them
- Please clarify the legend of Figure 3. Mention which model corresponds to which ECM

**Questions:**

1. The noisy case is not explained very well. Are the exogenous parents estimated based on an estimated model for the functional relationship?
The demo.ipynb seems to rely on additive form assumptions, which should be highlighted and discussed in the paper.

**Limitations:**

See Weaknesses

---

> ### Author Rebuttal · Authors · 2024-08-03
>
> We are glad you agree our method embeds the causal inference literature in a general framework for model explanations and that you found our paper relevant and well-supported.
>
> Thank you for pointing out the minor comments and typos. We will be careful to address each of these in the revision.
>
> We now respond to your question and the stated weakness together because they are linked. You are correct from your reading of the demo notebook that we use additive noise modeling assumptions. Our implementation also estimates the functions in the ECM using the explanatory dataset. In some examples we assume the (graph) structure of the ECM is known, but in the breast cancer data example we also learn the structure using the PC algorithm. So, **while you are correct that complete knowledge of the DAG and functions in an ECM is a strong assumption, we have actually already included work showing the cases where the functions are estimated and the structure learned from data**.
>
> The reason we define CDPs to take the ECM as an input is to make them modular. This way we are not wedded to any specific structure learning algorithm or function estimation method. A user can choose from the literature an estimation method that best fits with their application and then apply that method while constructing their ECM. **You correctly pointed out that we are building on other known results in the causal inference literature, so this actually makes CDPs more broadly applicable**.
>
> While our repository with code and notebooks shows implementation details, for clarity we will add more text description in the paper (e.g. how we constructed the ECMs), and also emphasize this point that other implementations could make use of different, context-specific learning algorithms and estimation methods, and then construct CDPs using their differently-estimated ECMs.
>
> Thanks again for your feedback and we hope this has answered your question. Let us know if you have any other concerns or suggestions on how we can improve our paper.

---

> > ### Comment · Reviewer_GtVe · 2024-08-09
> >
> > Thank you for addressing my comments.

---

> > > ### Author Response · Authors · 2024-08-12
> > >
> > > Thanks for acknowledging our response. The end of the discussion period is approaching but there is enough time for us to answer any additional questions if you ask them soon. We would also appreciate letting us know if you're willing to increase your rating since we addressed your comments.

---

> > > > ### Comment · Reviewer_GtVe · 2024-08-13
> > > >
> > > > Taking into account all reviews, I will maintain my score.

---

### Official Review · Reviewer_M8fQ · 2024-07-10

**Soundness:** 2
**Presentation:** 3
**Contribution:** 2
**Rating:** 6
**Confidence:** 3

**Summary:**

The authors present a novel set of attribution methods called Causal Dependence Plots (CDPs) that extend partial dependence plots (PDP) while respecting possible causal dependencies in the method's inputs. The approach aims to obtain truthful and reliable insights for black box model analyses. In detail, a given Explanatory Causal Model (ECM) is utilized to simulate the propagation of possible input intervention effects onto other causally related input variables. The authors claim that, by propagating causal effects for the inputs, a 'natural' configuration (according to the ECM) is presented to the model. This stands in contrast to existing attribution methods, which only vary features independently -- leading to out-of-distribution configurations that do not truthfully reflect the input domain of the model under consideration. The authors differentiate primarily between direct (or 'partial') dependence plots that only evaluates single input variation (corresponding to previous evaluation methods), and the total dependence, which consider changes in causally related inputs accordingly.

Evaluations are performed on multiple setups of synthetic toy examples and a real-world causal protein data set. In both settings, incorporating causal relations yields a better understanding of the system's behavior than non-causal attribution methods.

**Strengths:**

While the presented approach is seemingly simple, it has not yet been proposed to the best of my knowledge. Considering causal relations between input features when evaluating models is a clearly relevant and appealing idea that allows practitioners to consider either total or direct effects for their particular applications. The authors suggest applications for the important areas of evaluating fairness, distribution shifts, and theory testing.

The novel concepts are well presented and build upon each other. Additionally, the authors prove in Thm. 2.9, a particular form of CDP, NDDPs, are equal to PDPs with Individual Conditional Expectation (ICE).

Several variants of CDPs are presented and compared to each other. Experiments clearly exhibit the qualitative differences to previous PDPs and other attribution methods, such as Shapley values. The authors consider the cases of incomplete causal knowledge and perform a sensitivity analysis of their method.

**Weaknesses:**

(A) My main concerns are regarding the significance of this work. The presented contributions concerning causal inference seem rather minor, as the paper is mainly about the visualization of already-known results. In my opinion and the context of this conference, possible considerations and/or analysis about learning or causal inference are lacking.


(B) While the authors discuss possible important applications to fairness, out-of-distribution testing, or theory testing, none of the presented examples feature any of these topics.


(C) The authors primarily compare to associational methods. While not touching on the topic of visualization, causal Shapley values [Heskes et al. 2020] exist and should be compared to.


Minor remarks:
* On page 2, NDDPs are mentioned without being referenced or defined before.
* Sec 1. l.46 typo'containsing'



[Heskes et al. 2020], "Causal Shapley Values: Exploiting Causal Knowledge to Explain Individual Predictions of Complex Models"

**Questions:**

My questions mainly concern the mentioned weaknesses. I would, therefore, like the authors to comment on the following questions:

(1) Regarding (A): Could the authors think of possible applications of their method with causal analysis beyond pure visualization? When computing CDPs, how could a practitioner or automated system learn to infer treatment effects and detect deviations from the expected causal effects?

(2) Could the authors discuss possible relations of their work to [Heskes et al. 2020]?

(3) The authors mention the particular analysis of "causal descendants" (l.88). However, inference and predictions can also be performed in the anti-causal direction (consider, for example, [Schölkopf et al., 2012]). How does the presented approach translate to this kind of setting?



[Heskes et al. 2020], "Causal Shapley Values: Exploiting Causal Knowledge to Explain Individual Predictions of Complex Models"

[Schölkopf et al., 2012] "On Causal and Anticausal Learning"

**Limitations:**

By proposing a causal explainability method, various questions regarding the applications and risks of such methods arise for real-world decisions being made based on the resulting visualizations. The authors thoroughly discuss possible societal and ethical impacts due to improper use of their method. Implications of providing incorrect causal knowledge and a sensitivity analyses of the method are provided.

---

> ### Author Rebuttal · Authors · 2024-08-04
>
> Thanks for your review. It motivates us to make some small but key additions that should improve the paper. We take the claimed weaknesses and questions seriously and will do our best now to respond to each.
>
> A: Our main goal is not to contribute novel methods for causal learning or inference, but to reformulate (visual) model explanations as a causal problem within a clear conceptual and formal framework. As a comparison, consider the SHAP paper, "A Unified Approach to Interpreting Model Predictions" (Lundberg and Lee, NeurIPS 2017). This paper was not trying to contribute novel results or methods for game theory, but to reformulate model explanations as a game theoretic problem. Can such reformulations be significant? Perhaps- the SHAP paper has now been cited over 24k times. We are not conjecturing that CDPs will have a similar number of citations, but **work (like ours) which reformulates an important task (model explanation visualizations) as a causal one (and therefore connects it to much of the existing literature on causality) could end up being significant**.
>
> Q1: Consider the example in Section 2.7 (uncertainty ribbon plots) along with the final sentence of the conclusion:
>
> > Interpretability provided the initial motivation for CDPs, but since plots are qualitative CDPs also open the door for future work on causal methodology that relaxes assumptions while maintaining visual validity.
>
> These are some initial steps and **we hope to explore this direction more in future work**. Providing a novel estimation or inference procedure along with CDPs would require more theory and experiments, and possibly more discussion and related work. We think that the space constraints of one conference paper make it infeasible for us to include that exploration in the current paper.
>
> B: You are absolutely right. We can and will add examples that elaborate on these ideas, at least for the first two (which are more relevant to machine learning). Some of this may need to appear in the supplemental material, but if the paper is accepted we also have one additional page we can use to showcase an important motivating application like these. **We will write about how the fairness literature contains several causal formulations that may include direct and indirect discrimination, and that our different CDPs can be used to probe a model for such (un)fairness properties, while other methods only show direct discrimination**. Additionally, we will include an example where a model does not take a sensitive attribute as a direct input, but the ECM allows testing for that sensitive attribute’s influence on the prediction through its effect on the other inputs.
>
> C and Q2: We have found 3 papers which define some type of causal Shapley values, including the one you mention.
>
> - Causal Shapley (Heskes et al, 2020)
> - Asymmetric Shapley values (Frye et al, 2020)
> - do-Shapley (Jung et al, 2022)
>
> We have work in progress on comparisons with these. Among them, the do-Shapley method is closest in spirit to our approach and would likely make the most natural comparison. Unfortunately, so far we have had difficulty using the code from that paper’s supplement, and the same is true for the Causal Shapley paper. The ASV (asymmetric) paper is the only one with code that we’ve gotten to work by now, so we can most likely include an experiment to compare with that. We will continue working on the others and hopefully be able to include them all, but cannot promise that. At minimum, **we will include more discussion about the Heskes et al paper as requested**, and the other two, as related work. While we are working hard to include more comparison with these works, it is also worth mentioning that their goals differ somewhat from ours. They do relatively simple and automatic feature attribution without any specific guiding or motivating question. A strength of CDPs is that they can be constructed with an arbitrary intervention, and that could be motivated by a specific model auditing or scientific question.
>
> Q3: There are two near connections between our work and anti-causal learning. The first is that, as in anti-causal learning, the case where a cause is predicted from its effects has special implications for CDPs. CDPs show the causal dependency of the model’s predictions on a given variable, and in this case the direction of causality appears reversed from the underlying true outcome. This is another example about why explaining a model and explaining the DGP can be fundamentally different questions. The second connection is that a type of semi-supervised learning could be useful in both settings. For CDPs, unlabeled data could possibly be used when constructing an ECM. **We will expand our discussion of these points in the paper**, either with the additional page after acceptance or in the supplement.
>
> You pointed out that our proposal is simple, novel, appealing, relevant, well-presented, and clearly shown to be qualitatively different from the existing comparison methods. We thank you for this assessment, as well as your more critical feedback. Please let us know if you have any other questions, suggestions, or concerns.

---

> > ### Comment · Reviewer_M8fQ · 2024-08-10
> >
> > Dear authors,
> > thank you for answering my questions.
> >
> > Q2: I am not sure whether direct comparisons make sense here, as the related papers put different focus on different aspects. I was mainly concerned with the overall lack of discussion on related methods and believe that this point has been cleared.
> >
> > Q3: I agree with the authors' view on the topic. Thank you for providing further insights and including a discussion in your paper.
> >
> > Regarding Q1: While I definitely agree that visualization/explanation methods are relevant contributions to the field, I also believe that the paper is still lacking behind in its possibilities. In particular, if one is already in the position of having access to a causal model, it also enables us to attribute or trace-back model misbehavior to particular factors in the SCM (e.g., identifying a particular variable being ignored by the predictor, thus being the cause of its mispredictions). For now, the paper only utilizes causal models to adjust input data, but does not leverage their ability of inferring explanations using the strong computational/structural implications of an SCM.
> >
> >
> > I have raised my score to borderline accept, but would still like to encourage the authors to think about whether such an analysis could be applied to any of the existing experiments, or being discussed in the context of some simple artificial example.
> >
> > Best,
> > Reviewer M8fQ

---

> > > ### Author Response · Authors · 2024-08-10
> > >
> > > We're grateful that you increased your score and replied to our rebuttal.
> > >
> > > Your follow-up on Q1 raises an excellent point. It is actually something we thought about as another potential future application of CDPs: **diagnostic plots of residuals**. We didn't mention it in this paper because we were pressed for space, and also because we thought the topic was less well-known.
> > >
> > > We don't assume the explanatory dataset is labeled or that the ECM includes the outcome variable. However, if it does, then we can plot the residuals on the vertical axis instead of the predictions. This could be useful for exactly the type of example you mentioned.
> > >
> > > We are willing to add a brief discussion about this and possibly include another plot, for example showing the residuals from the random forest and/or linear models from Figure 1. This might need to go in supplemental material depending on the space remaining. Let us know if you think that would be a valuable addition to the current draft- it would not take much additional work, so it would be another minor change we can add to our action items.
> > >
> > > Thanks again for your reply and for raising this point.

---

> > > > ### Author Response · Authors · 2024-08-12
> > > >
> > > > Dear Reviewer M8fQ,
> > > >
> > > > There is still a little time left to discuss our reply to your last point about Q1, that is, about including an example showing the use of diagnostic plots of residuals (which could potentially be used to find problems with the predictive model and/or the ECM). Is this along the lines of what you were asking? Would it help address your Q1 sufficiently to merit any more of an increase in your rating? Do you have any clarifying questions or remaining reservations?
> > > >
> > > > Best,
> > > > The Authors

---

> ### Comment · Reviewer_M8fQ · 2024-08-12
>
> Dear Authors,
> thank you for again addressing my concerns regarding Q1. While residual plotting is a delicate matter on its own, --due to mispredictions possibly not becoming apparent at the root cause of the misprediction, but only later on, due to possible cancellation/reinforcing effects--, I would appreciate the demonstration of such an analysis in the paper or appendix.
>
> Given that the claimed additions towards my remarked points will appear in the final version of the paper, I'm willing to increase my score to a weak accept.
>
> Best,
> Reviewer M8fQ

---

### Official Review · Reviewer_o4Ea · 2024-07-12

**Soundness:** 3
**Presentation:** 4
**Contribution:** 3
**Rating:** 6
**Confidence:** 5

**Summary:**

The authors propose a new approach to visualize the impact of a change in a variable on the outcome of interest. They argue that if some of the other variables in the model are mediators, and that they should not be held constant as is currently done in variable importance measures, as it may lead to bias (post-treatment bias) on the measure of the effect on the outcome. They define estimands derived from the field of causal mediation analysis, and show how to use them in graphical representations that are easy to interpret and can also incorporate uncertainty.

**Strengths:**

- the authors have identified a flaw in the field of explainable ML and propose intuitive ways to remedy to this issue
- the proposed graphical representations are very informative and useful
- the authors propose to introduce more causal reasoning into the model explainability, while doing so with easy to interpret graphical representations. I have just updated my rating as I had misjudged the field of the paper, which made it a little weak from a causal inference perspective, but rather original and useful in the explainable AI field

**Weaknesses:**

- this work is not very theoretical, but it introduces a new way to look at a model results for explainability
- however, it is a small fix on the problem of causally interpret machine learning models instead of doing a complete causal analysis. The PDP approach proposed by the authors require the same assumptions as a causal analysis: definition of the estimand of interest, verification of the identifiability assumptions, possibly through the construction of a causal graph, and estimation. From Figure 1, we can see that the results are sensitive to the choice of mode, and no guidance is provided on how to chose a suitable model. Overall the authors propose a fix for users that want to interpret a machine learning causally. However, if the model does not include a suitable set of adjustment variables, this approach will not work. Maybe a clearer message would be to to causal inference when one wants a causal interpretation of the model, instead of tinkering a non-working approach. However, the provided plots are interesting, and should be used in mediation analysis.

**Questions:**

- Can you provide clearer guidance for the user in choosing an adequate model?
- and for users to correctly verify the assumptions are met, especially if they are not familiar with causal inference and come from the field of explainable AI

**Limitations:**

the idea is good, but probably hard to use in practice, especially for users that are not familiar with causal inference. It also encourages users to think that they can obtain a valid causal interpretation from an additional step after fitting a machine learning model, instead of insisting that a valid causal conclusion is prepared by all steps of the analysis, including the study design, which is crucial.

---

> ### Author Rebuttal · Authors · 2024-08-03
>
> Thanks for your assessment. We are happy you found the soundness and contribution of our paper good and the presentation excellent. We agree that most explainable ML has a serious flaw and that an approach based on causal graphs and visualizations is intuitive, useful, and very informative.
>
> We also agree about the importance of study design, particularly thinking about the choice of estimand, requirements for valid estimation and inference, and, ideally, doing this all before collecting data or fitting models. It is unfortunate that good statistical practices like these are too rarely followed. In the domain of NeurIPS (ML/AI), it is common for people to have datasets with plenty of black-box predictive models already fit to them, and only after achieving SOTA validation accuracy do they start to question why and how the model "works." It is also common for people to use other popular model explanation / feature attribution tools, like SHAP values, and (incorrectly) interpret the results causally. **That is the status quo, that is what our work is trying to improve, and we believe that providing tools like CDPs will point such users in a better direction**.
>
> After our points of agreement there are a few issues we must push back on.
>
> Firstly, it is outside the scope of our one paper to provide general guidance on choosing a good causal model. That is the subject of much other work, and it will take a lot of time and education reforms for good practices from this field to make their way into common use. Again, we believe that CDPs will help in these efforts by pointing users who wish to explain/interpret an ML/AI model in the direction of causality. Causality is a large topic, and such users will have to inform themselves by reading other papers and books. **We are very clear in the paper that the limitations of identifying a good causal model apply to using CDPs**. This is responsible, this cannot be left to other work, and we have done it in Figure 1 and in the discussion. In the revision **we will add more references to resources in the causal literature that can provide general guidance to readers**.
>
> Secondly, we disagree that CDPs "[encourage] users to think that they can obtain a valid causal interpretation from an additional step after fitting a machine learning model." On the contrary, **we repeatedly state that interpreting the output of a black-box model is fundamentally different from causal inference about the real world outcome variable**. See the third takeaway point about Figure 1 (starting on line 75), the second paragraph of Section 3 (Experiments), and lines 297-98 in the discussion (Limitations). We also define ECMs (Definition 2.5) with a convention of representing the predicted outcome as a separate node in the ECM graph, distinct from the true outcome variable, and directly caused by each predictor input of the black-box model.
>
> Lastly, and perhaps less importantly, the review overemphasizes the importance of mediation. The parts of our paper that focus on simple mediation examples (the introduction and Section 2.6) are included mainly to help understand the basic idea. But with Partially Controlled Dependence Plots (PCDPs) and the direct dependence of model output on all predictors, CDPs are distinct from and more general than typical mediation analysis. **We will emphasize this by including one more definition of a general CDP using an arbitrary family of interventions parameterized by the plot axis** (e.g. it can be an intervention on multiple predictors simultaneously, for example subtracting from one and adding to another, or more generally moving in a certain direction in the input space).
>
> Your review makes us think you are very knowledgeable about causal inference. We are curious to know what you think about the last line of our conclusion:
>
> > Interpretability provided the initial motivation for CDPs, but since plots are qualitative CDPs also open the door for future work on causal methodology that relaxes assumptions while maintaining visual validity.
>
> In other words, we think the assumptions required for valid quantitative inference (e.g. a confidence interval) are more restrictive than those that will be necessary for some notion of good qualitative inference (e.g. something about the overall shape of the plot being correct). We think this is an exciting direction for more work in causal inference generally once CDPs have been established.
>
> Thanks again for the feedback. Your assessment was overall quite positive, including the soundness, presentation, and contribution scores. Please let us know if you have other questions or suggestions.

---

> > ### Comment · Reviewer_o4Ea · 2024-08-09
> >
> > I thank you for your answers and clarifications, I agree that you are clear about the distinction of the approach you propose and a causal inference approach, however, I think that the difficult part of causal inference is establishing a reasonable DAG and ECM, which is necessary for the CDP, but does not provide a result as strong as a causal result.
> >
> > Regarding your final remark, CDPs will definitely provide valuable insights, however, some dependencies can be reversed if some relevant variables (confounders or others) are missing from the model, or if other variables (mediators or colliders) are adjusted for, and it is not (yet) clear to me how CDPs can overcome those limitations.
> >
> > Providing the rephrasing and explanations and references that you are considering to include in your work I am willing to change my grade to 6 (weak accept).

---

> ### Author Response · Authors · 2024-08-09
>
> Thank you for reading and responding. We're pleased to hear you are willing to increase your score.
>
> If you have any particular references that you recommend for general guidance on causal modeling let us know so we can consider including them.
>
> Two brief comments follow just so we can be sure we're understanding each other.
>
> > [...] the difficult part of causal inference is establishing a reasonable DAG and ECM [...]
>
> We agree with this, but we also think it is not necessarily a weakness about our paper. Our task and contribution is not a method for causal inference but for model interpretation, that is why the paper is submitted under the "**Primary Area**: Interpretability and explainability." For reasons that model interpretation itself is an important task, we'll refer again to the potential Applications listed in Section 1.1 of the paper, e.g. "multi-party auditing."
>
> There are also some reasons we think the ECM requirement is not too strong, e.g. the paragraph on "incomplete causal knowledge" and Section 2.7, which brings us to the next comment.
>
> > [...] it is not (yet) clear to me how CDPs can overcome those limitations.
>
> It's fair that this is not entirely clear yet, at this stage we have only demonstrated some proofs of concept. What we are thinking about here is something like the example in Section 2.7. We think the problems that arise due to the challenges you mention, like whether some mediator/collider is adjusted for or not, might turn out to be less problematic when we look at their impact *on a plot*. In other words, the assumptions for a certain estimator to have a certain desirable property (e.g. double robustness) might fail to hold, but a plot with an accompanying uncertainty region might show the empirical relationship is qualitatively similar (e.g. increasing but concave) across a large range of conditions. If that qualitative visual conclusion is what's important for the given application (e.g. a paper testing some theory which predicts the relationship should be increasing but concave) then we don't need to worry about the other assumptions that were failed.

---

> > ### Author Response · Authors · 2024-08-12
> >
> > Since the discussion period will end soon we wanted to ask one last time if you have any more follow-up questions or comments.
> >
> > You previously mentioned you were willing to increase your rating to 6, and we hope our last reply may have improved your opinion more. We discussed a similar point with Reviewer M8fQ:
> >
> > > Our main goal is not to contribute novel methods for causal learning or inference, but to reformulate (visual) model explanations as a causal problem within a clear conceptual and formal framework. As a comparison, consider the SHAP paper, "A Unified Approach to Interpreting Model Predictions" (Lundberg and Lee, NeurIPS 2017). This paper was not trying to contribute novel results or methods for game theory, but to reformulate model explanations as a game theoretic problem. Can such reformulations be significant? Perhaps- the SHAP paper has now been cited over 24k times. We are not conjecturing that CDPs will have a similar number of citations, but work (like ours) which reformulates an important task (model explanation visualizations) as a causal one (and therefore connects it to much of the existing literature on causality) could end up being significant.
> >
> > Whatever number you are willing to increase your rating to, if you update the form before the discussion ends then we would be able to see the change, and we would be grateful for that.

---

> > > ### Comment · Reviewer_o4Ea · 2024-08-12
> > >
> > > I have updated my review, and the rate.

---

### Official Review · Reviewer_6pua · 2024-07-15

**Soundness:** 3
**Presentation:** 2
**Contribution:** 2
**Rating:** 6
**Confidence:** 4

**Summary:**

The authors tackle the problem of evaluating the dependence between the inputs and output of black box machine learning models.  Generally, analysis of this type of dependence is done in a univariate manner, holding constant all but one variable and visualizing how the outcome changes as we vary that one variable.  However, when the variable of interest is causally related to other input variables, these results may be misleading and only show part of the picture.  To help elucidate the bigger picture in these cases, the authors propose a framework that generalizes partial dependence plots, allowing for the visualization of the dependence between input variables of interest and the outcome in terms of metrics like total, direct, and indirect effects.  The authors demonstrate this approach on simulated and empirical data.

**Strengths:**

I really appreciate what this paper is trying to do.  Explainability of black box models is only getting more important, and providing easy-to-use visualization tools can be invaluable when trying to understand the uses and limitations of these models and how they can be used in practice.  The issue of univariate explanations and what to hold constant is described quite well, helping to lay a strong motivational basis for the authors' work.  I also think the writing is generally clear, and I appreciate the authors putting a motivating example very early on.

**Weaknesses:**

I really want to like this paper.  I love visualization tools and explainability, so work like this appeals to me.  However, I feel like the way this work is presented does it a disservice.  For a paper introducing a new visualization and explainability approach to be adopted, readers need a clear picture of how to apply these types of visualizations to their problems, what sort of results they can expect, how to interpret those results, and then what to do with them.  The authors do provide an example early on in Figure 1 with some key takeaways and some brief discussion of results in the Experiments section.  However, the discussion is generally both too high-level (such as saying "this plot shows X" without clarifying how to read X of the plot) and stops short of where actual application would need to go (such as saying "we can see from the plots that X and Y differ", without describing why that is significant and how a practitioner should interpret that difference).  This results in a paper that seems to have some interesting visualization ideas and methods but that I feel would struggle to see the methods actually adopted in practice without clearer application guidance.  I struggled a lot with what score to give, and I'm absolutely open to changing it based on other reviewers' comments and the authors' response, but for now, I can't quite vote for accept given these issues.

More specifics:

Figure 1 is close to being very helpful as an early intuition-builder, and, with a little bit more description, could actually be very useful.  I think my main issues with it as-is are:
- The left 3 plots have the orange and blue lines as defined in the figure description, but the description doesn't say what the light grey points are in the plot.
- The caption describes the orange lines as "Natural Direct Dependence", a term that is not actually defined until page 7.  The text does, at least, describe the orange lines as the dependence "when F is held constant at its observed value", but then describe it as coinciding "exactly with a standard PDP", another term which is not defined outside the appendix.
- I actually really like the three points of takeaways from Figure 1.  However, at least for me, they're missing a step: what exactly are you seeing in the plots that leads to these takeaways?  For example, the first takeaway is that "there can be qualitative differences between direct (or partial) dependence and total dependence."  However, given that we're still in the introduction and these terms haven't been clearly defined yet, you should clarify that you're concluding this by comparing the blue and orange within each individual plot and ideally also what this means semantically for the example. (i.e., If we just looked at total dependence, we may conclude X, but this plot shows that partial dependence actually behaves like Y) The third takeaway is also mostly a great description, but I think I'm still missing something.  Does the takeaway that random forest messes up on direct dependence mean that the random forest model is incorrect?  If all I had done was train a random forest model and saw that result, would I have any way of knowing that it's incorrect?

Even as someone who knows what counterfactuals are, I found your definition of counterfactuals in Definition 2.3 confusing.  It sounds like we're reasoning about interventions on $V_j$, and this intervention could be setting it to a constant or defining a new function for its dependence on its parents. (based on Definition 2.2) However, Definition 2.3 also describes the possibility of "also do[ing] an intervention that changes any of the values in $PA_j$".  I'm also not sure what it means that you "may hold some or all of $v$ fixed vary $U_j := u$" - if we're varying $U_j$, does that mean the intervention is touching all of the exogenous parents of $V_j$ but maybe only some of the observable parents? But I don't see how the counterfactual definition requires us to vary the parents - if we're intervening at $V_j$, it shouldn't affect its parents, right?  Just how the values of those parents affect $V_j$.  There must be something I'm missing in this definition. (I also don't think it helps that you use capital $V$ for the variable of interest but lowercase $v$ for its parents)

Section 2.4 has some symbology/variables that don't seem well-defined.  I think line 176 is the only place the symbol $\mapsto$ appears (unless I'm missing something).  What does it mean here? Also, $k$ suddenly appears here as an index to $g$ and $x$, and we see that it ranges from $1$ to $p$.  Given how important this section is, a little more clarity in all of these indices would help.

Algorithm 1 seems strange and unnecessary.  First, on a bit of a side note, line 195 says that Algorithm 1 describes "the construction of an ECM", which seems like an odd title.  Up until now, the ECM has been discussed as the causal model provided by the user, and all algorithm 1 seems to be doing is adding edges from every variable in the ECM to the outcome, so I don't really see how that can be called "constructing the ECM."  More importantly, though, Algorithm 1 seems essentially just like a pre-processing step.  You could just as easily tell the user to provide an ECM describing the causal structure among the input variables that has all of them causing outcome. (which would probably be the default assumption anyway, hence why they're being included as predictors) Given how tight space seems to be, unless there's a nuance I'm missing, Algorithm 1 could just be replaced with the sentence "First, we add edges from all variables in the ECM to outcome."

Line 196 introduces the notation of $\hat{f}(P^M)$.  However, this notation was used for the first time on line 191 - introduce this notation before you use it.

In Algorithm 2, the second line is "Get the possible values of $X_S$ and set it to $X$".  I think you mean the opposite (i.e., "Set $X$ to all the possible values of $X_S$), since I'm not otherwise sure what it would mean to set all the possible values of $X_S$ to $X$...

I appreciate the discussion in 2.7 about uncertainty of the ECM structure, but, as with the discussion around Figure 1, I think it stops just short of providing a strong enough example.  If I were uncertain about the structure and produced the plots in Figure 3, what should be my takeaway besides "the shapes are different for each model"?  I did look at the example in Appendix B.2, hoping that the added space would allow for this type of discussion, but the only written takeaway for Figure 8 is "cell shape impacting tumor class is indeed sensitive to our choice about the uncertain edge, particularly for the TDP."  The difference does appear very large for TDP, so what does that mean?  Can I use these results in any way to figure out which MEC member I should use?  Is the takeaway "when reporting these results, I should report values for all three models?"

In line 273, the authors conclude that, since ALE and SHAP appear similar to the PDP, "our TDPs represent a significant and novel contribution to the existing model visualizations."  I don't necessarily disagree that the authors' TDPs are useful, but I just don't see how that follows from the previous statement.

The authors appear to be very pressed for space, relegating a lot of content to the supplementary material.  However, while some of this is reasonable and understandable, there are places where this is either confusing of feels clunky.
- The authors describe their work as a generalization of PDPs.  However, the authors never actually define PDPs in the body of the paper, leaving that for the appendix.
- Some of the discussion on page 7 seems overly dependent on the supplementary material, to the point where it's not particularly clear or useful without it.  This especially stood out in lines 224 and 225, where we're asked to compare Definition 2.8 to a definition in A.2, which shows that it's equivalent to the PDP (which is defined in A.1), the proof of which is in A.2.  Rather than having this content awkwardly split between the main paper and the appendix, I'd rather have this stuff referenced at a higher level with a pointer to the appendix, freeing up some space in the main paper for more detail on other parts that need it.

**Questions:**

I think these are mostly covered by the Weaknesses section.

**Limitations:**

The biggest weakness of this approach is the need to specify the ECM structure ahead of time.  To the authors' credit, they are upfront about this and provide multiple ways to ease this challenge, such as using causal structure learning methods (such as in the example in the appendix) or by allowing the user to supply multiple candidate models.


Based on the authors' responses, I am raising my score from a 4 to a 6.

---

> ### Author Rebuttal · Authors · 2024-08-01
>
> Thanks very much for your insightful and thorough review. You’ve given us useful feedback that will help clarify and improve the paper. You’re right that we were pressed for space- the CDP framework is quite general and there are many things we would like to demonstrate with it (e.g. other interesting special cases of PCDPs for common DAG structures, more about uncertainty regions, applications to fairness, etc). Accepted papers will have one additional page in the main text, and we can recover some space by removing Algorithm 1 as you suggest. Using that space and this response, we will do our best to answer all of your questions. We hope that you end up liking the paper as much as you wanted to!
>
> On Figure 1 and the introduction:
>
> - The light gray points show the explanatory dataset (the same dataset used by the comparison methods)
> - **We will add a pseudo-algorithm description of PDPs in the introduction**. This will help serve as a contrast for our proposal, making it clear that the difference is PDPs holding other variables constant. While the NDDP is not defined until later, this should still be enough to drive home the first main point that TDP and PDP are fundamentally different.
> - We will describe how the takeaways are reached from the plot with sentences like the following. On qualitative difference: "Across multiple black-box models the TDP shows an overall stronger increasing total dependence compared to direct dependence. And the sign can even be flipped, as in the true DGP and the linear model where the direct dependence is negative while the total dependence is mostly positive." On the third point: "The direct dependence of $\hat S$ on $P$ is increasing while the direct dependence of $S$ on $P$ in the true DGP is decreasing. In this case studying the black-box would not necessarily help us learn about the true DGP."
>
> For Definition 2.3 **we will clarify that we are not intervening on $V_j$, but describing its counterfactual value under some intervention that modifies other variables** (some of which may be observed or exogenous parents of $V_j$). We need this level of abstraction for partially controlled effects, and that may make our definition less similar to cases where the application does not require partial control.
>
> In Section 2.4 **we will remove the $\mapsto$ symbol and better explain the indices**, describing the overall point in the text: "All the other variables $x_k$ are uniquely determined by variable $x_j$ in this model."
>
> We will fix the out-of-order notation usage, thanks for pointing it out.
>
> For Algorithm 2 you are right and we will change the text (it depends on if we think of "set" as operating to the left or right, but leftward is certainly the most common)
>
> For Section 2.7, the most important takeaway is just that uncertainty about an ECM can be represented visually in CDPs. There are various ways to model uncertainty, and CDPs are agnostic/modular regarding that choice. We demonstrate one method based on having two (or a set of) ECMs. **We will add more text to expand on the specific takeaway**, like: "If we are not certain which of these two ECMs to use, this plot shows a region interpolating between both of their TDPs (or NIDPs). This example is not a confidence region, but any method for producing confidence sets in SCMs could also be used with CDPs to display uncertainty regions."
>
> > Is the takeaway "when reporting these results, I should report values for all three models?"
>
> We think best practices in applications could involve things like pre-registering which results will be computed/reported, though this is somewhat outside of our scope. It will depend on publication norms and requirements that vary between journals and fields.
>
> On the comparison with ALE and SHAP: Several **popular/SOTA xAI/interpretability tools produce qualitatively similar plots, while our TDP is the only one that stands out in showing a more sharply increasing relationship**. We want to conjecture that this generalizes: most of the xAI tools only show direct dependence. The theorem about PDPs is the only theoretical result we have been able to prove about this so far. Perhaps the general conjecture requires auxiliary hypotheses that vary between different application settings. A follow-up paper focusing on applications to fairness, for example, could add a condition that is likely to hold across many fairness settings, and then possibly prove (or show in experiments) that the other methods only show direct dependence (i.e. "direct discrimination").
>
> Finally, we reiterate that with the additional page for accepted papers we can move some supplemental material to the main text. **We will move the result showing PDP + ICE = NDDP to the main text**. Along with defining PDPs in the introduction, these changes will allow future readers to appreciate the main results and takeaways with less difficulty.
>
> Thanks again for your time and work. We sincerely appreciate your help improving our paper.

---

> > ### Author Response · Authors · 2024-08-12
> >
> > Your review was very thorough and we tried to match that with our response. We hope you found the response satisfactory, perhaps enough to merit increasing your rating. But if you still have follow-up questions or any remaining concerns we kindly request you inform us soon so there is enough time to respond.
> >
> > Thanks!

---

> > > ### Comment · Reviewer_6pua · 2024-08-13
> > >
> > > I appreciate the authors' detailed response, and i apologize for the late response.  I think with the proposed changes, this paper will be much more understandable and applicable, so I am happy to raise my score.

---

> > > > ### Author Response · Authors · 2024-08-14
> > > >
> > > > Thank you for replying and adjusting your rating based on the exchange. No worries about the timing, it would only have been difficult if you had additional questions that took substantial time to answer.

---

### Author Rebuttal · Authors · 2024-08-07

We are really grateful for these high quality reviews. We condense reviews/rebuttals below and invite reviewers to correct us if we changed the meaning or missed important points (full reviews and responses are of course available separately).

# Summary

## Reviewer 6pua

**Positives**: "I really appreciate what this paper is trying to do. [The problem] is only getting more important, and providing easy-to-use visualization tools can be invaluable [...]" and "The issue of univariate explanations and what to hold constant is described quite well, helping to lay a strong motivational basis for the authors' work. I also think the writing is generally clear." The paper is upfront about its most important limitation (requiring a causal model as an input) and also provides multiple ways to deal with it.

**Criticisms**: Some of the presentation is unclear. Several things are close to being helpful but need additional clarification. Some notation or definitions are confusing and/or presented out of order. Algorithm 1 is unnecessary. The paper is pressed for space and some important results have been put in the supplement but should be in the main text.

**Response**: With the additional page allowed for accepted papers we could move key material from the supplement to the main text. We will clarify the identified definitions/notation and expand some descriptions/explanations.

## Reviewer o4Ea

**Positives**: Identifies a flaw in ML explanations and proposes an intuitive solution. "[V]ery informative and useful" with high scores on soundness (3), presentation (4), and contribution (3).

**Criticisms**: The work is not very theoretical. It provides only a small fix that will encourage users to interpret a machine learning model with a possibly wrong causal conclusion instead of properly verifying the assumptions required for correct causal conclusions. There should be more guidance for users on how to choose a good causal model.

**Response**: These criticisms are most appropriate when the motivating question is to make causal conclusions about the underlying data generating process. But CDPs are, firstly, a tool for explaining a predictive model. It is true that additional assumptions are required for an explanation of a predictive model to have any validity for the underlying true outcome, but that is only one specific application of model explanations. The CDP framework helps distinguish between these applications and makes it more clear that causal conclusions depend on assumptions that must be checked. We will add more references to other important causal literature that readers should be familiar with in order to use CDPs.

## Reviewer M8fQ

**Positives**: The appoach is simple, novel, appealing, relevant, well-presented, and clearly shown to be qualitatively different from the existing comparison methods. Evaluation experiments in multiple synthetic and real datasets show "incorporating causal relations yields a better understanding of the system's behavior [...]." Limitations are thoroughly discussed. Included sensitivity analysis for the case of incomplete causal knowledge.

**Criticisms**: The contribution for causal inference is minor, lacking analysis or guidance. Possible applications (e.g. to fairness) are mentioned but not demonstrated in any examples. There should be some discussion about an existing method for causal Shapley values and an existing paper about anti-causal learning.

**Response**: Our contribution is not targeted to causal inference but rather to explainable/interpretable ML/AI. As an example for comparison, the SHAP paper was not a novel contribution on game theory, but reformulated feature attribution as a game theoretic problem. We aim to do a similar thing with feature visualization. We will add more detailed explanation and/or examples for applications like fairness and distribution shift. We will include more discussion about recent work on causal Shapley values and the connections between our work and anti-causal learning.

## Reviewer GtVe

**Positives**: New general framework for model explanation plots that leverages/builds on existing literature on causal inference. Claims are well-supported, ideas and results easy to follow, topic is relevant.

**Criticisms**: CDPs use a fully specified ECM which is a strong assumption. Paper should discuss the case if ECM functions are estimated.

**Response**: We have already shown examples where the functions are estimated and even when the structure is learned from data. In Section 2.7/Figure 3 and Section B.2/Figure 8 we have shown a way to visualize uncertainty coming from that step. And as the Reviewer correctly points out, since we provide a general connection/bridge to the literature on causal inference, users can adapt any application-specific method for causal estimation and inference to construct their ECM and visualize uncertainty regions.

# Our action items

Some specific concerns were raised which we're confident can be addressed with the minor revisions below. (Accepted papers get 1 additional page)

- Add defn/pseudo-code for PDP in intro
- Move PDP + ICE = NDDP to main text
- Condense Algorithm 1
- Small change of notation and definitions to emphasize that predictors could be a subset of ECM variables (an intervention could target something which is not a direct input to the model)
- Add most general CDP: any family of interventions parametrized by the plot axis
- More about fairness, distribution shift, causal SHAPs, connections with anti-causal learning
- More refs to other causal literature with guidance
- More detail about experiments (how we fit ECMs)
- Fix some typos

# Discussion

The reviews are of very high quality and fairly positive overall. Even those which were more critical or gave lower ratings still said strong positive things about the paper. Hopefully reviewers find we've engaged their writing substantively and answered their concerns. We look forward to a productive discussion

---

> ### Author Response · Authors · 2024-08-14
> **Update after discussion**
>
> Thanks to the Reviewers and AC for a responsive and productive discussion. Here are a few updates about our action items:
>
> - We've found another causal SHAP paper (Counterfactual SHAP, Albini 2022) and will add it to our discussion of related literature. Reviewer M8fQ also clarified that we are only requested to include discussion of causal SHAP, and not necessarily experimental comparisons (since we noted it has been difficult to use some of the existing code implementations)
> - Following more discussion with Reviewer M8fQ we will add some discussion about diagnostic plots of residuals, and potentially an experiment that illustrates the basic idea (possibly in an appendix)
>
> Each of the proposed changes is manageably small and together they will improve the existing paper without changing its key contributions.
>
> We're grateful for all the feedback so far, and will listen if there are any more recommendations accompanying the final decisions.

---

### Comment · Area_Chair_vcrX · 2024-08-07
**The discussion has opened**

Dear reviewers,
thank you so much for the precious work you have done so far. Your comments are fundamental to help the authors of the paper to improve the quality of their manuscript. The authors well recognized the great work you have done so far as well as I strongly agree with them.
The authors too seriously your comments and prepared rebuttals which are well structured and meticolous, thus I kindly ask you all to read them carefully and to discuss directly with author whether their answers to your comments are satisfactory or not.
All the best

---

### Comment · Area_Chair_vcrX · 2024-08-10
**An invitation to share your opinion on rebuttals**

Dear reviewers,
I kindly ask to those of you that did not yet share your opinion on rebuttals to do as such to help tha authors to improve the quality of their work.
All the best

---

### Comment · Area_Chair_vcrX · 2024-08-12
**Urgent need to share your opinion on authors' rebuttals**

Dear reviewers, in particular those of you that did not commented on the authors rebuttals.
Please share your opinion on what the authors wrote in response to your reviews.
This is an urgent issue for this paper
All the best

---

### Decision · Program_Chairs · 2024-09-25

**Decision:**

Accept (poster)

**Comment:**

The consensus is full among the four reviewers, they awarded exactly the same rating at the first review.

The authors-reviewers engagement was high with 33 posts published on this manuscript.

The strenghts of the paper overwhelm the weaknesses.

I greatly appreciated the opinion from 6pua who raised several issues on how the paper is presented. In particular, a paper introducing a new visualization and explainability approach should give a clear picture of how the new technique must be applied, which kind of results can be expected and how these results can re interpreted and leveraged on. Another weakness is the limitation implied by the need to have a fully specified ECM. However, the rebuttal and subsequent discussion clarified on this aspect that was finally considered not that a strong limitation.

Amongst the strengths, I feel that the following are of some interest; i) the authors revealed a flaw in explainable ML while proposing a solution, ii)  the manuscript introduces more causal reasoning into the model explainability, while doing so they try to improve the interpretation of graphical representations. I share the same view as two reviewers that this manuscript is a little weak from a causal inference perspective, but quite original and useful in the explainable AI field. An additional strength of the paper is that it presents the novel concepts very well and also develops theoretical arguments (Theorem. 2.9),that are useful.

In conclusion, I think the paper deserves attention even if it could be significantly improved whether the authors can address and solve the issues raised by some of the reviewers. In particular, I think that to improve the quality of the paper the authors can refer to the excellent review from 6pua.